# Selected Acoustic Frequencies Have a Positive Impact on Behavioural and Physiological Welfare Indicators in Thoroughbred Racehorses

**DOI:** 10.3390/ani13182970

**Published:** 2023-09-20

**Authors:** Léa Gueguen, Séverine Henry, Maëlle Delbos, Alban Lemasson, Martine Hausberger

**Affiliations:** 1Univ Rennes, Normandie Univ, CNRS, EthoS (Éthologie Animale et Humaine)—UMR 6552, 35000 Rennes, France; severine.henry@univ-rennes1.fr (S.H.); 1maelle3@gmail.com (M.D.); alban.lemasson@univ-rennes1.fr (A.L.); martine.hausberger@cnrs.fr (M.H.); 2UMR 8002 Integrative Neuroscience and Cognition Center, CNRS, Université Paris-Cité, 75006 Paris, France; 3Institut Universitaire de France, 75005 Paris, France

**Keywords:** horse, welfare, racing, acoustic stimulation, EEG wave playback, musicotherapy, 432 Hz tuning

## Abstract

**Simple Summary:**

Racehorses are submitted to stress-inducing practices that include restricted conditions of life and intensive racing training. This may impair their welfare and, as a consequence, their rider’s safety. Horses with compromised welfare may express stereotypic behaviors, aggressiveness or apathy, and may present abnormal hematological data. Agitation and lack of sleep may also lead to difficulties in physical recovery. Acoustic stimulation has been proposed as a sensory enrichment for a variety of domestic species, but there are debates about what type of sound is best. In humans, it has been argued that particular frequencies could have beneficial effects on health. In the present study performed on 12 thoroughbred racehorses in training, we found that the daily playback for three weeks of a stimulus involving an array of these different frequencies was associated with a decrease in stereotypic behaviors and agitation behaviors and an increase in recumbency and hay ingestion. There was also an improvement in red blood cell-related parameters. Most of these effects lasted or still increased after the cessation of the playback phase. Overall, the animals appeared quieter and potentially experienced a better physical recovery.

**Abstract:**

(1) Background: Since antiquity, it is considered that sounds influence human emotional states and health. Acoustic enrichment has also been proposed for domestic animals. However, in both humans and animals, effects vary according to the type of sound. Human studies suggest that frequencies, more than melodies, play a key role. Low and high frequencies, music tuning frequency and even EEG slow waves used for ‘neurofeedback’ produce effects. (2) Methods: We tested the possible impact of such pure frequencies on racehorses’ behavior and physiology. A commercial non-audible acoustic stimulus, composed of an array of the above-mentioned frequencies, was broadcasted twice daily and for three weeks to 12 thoroughbred horses in their home stall. (3) Results: The results show a decrease in stereotypic behaviors and other indicators such as yawning or vacuum chewing, an increase in the time spent in recumbent resting and foraging, and better hematological measures during and after the playback phase for 4 of the 10 physiological parameters measured. (4) Conclusions: These results open new lines of research on possible ways of alleviating the stress related to housing and training conditions in racehorses and of improving physical recovery.

## 1. Introduction

Since Plato, in ancient times, it has been generally admitted that sounds, especially music, have an effect on human emotional state and health [1,2]. Nowadays, there are many reports on the different effects of music on human mental states such as relaxation/pleasure but also, according to music characteristics, increased agitation, as well as on physiological/health measures such as heart rate variability [3]; pain perception [4]; stress and immunity [5]; depression [6]. Music therapy has become an official branch of medicine in various countries, although it has not yet been well studied [1]. In animals, auditory stimulation has been proposed as a possible sensory enrichment for captive and/or domestic animals [7] to alleviate the effects of restricted living conditions and their impact on welfare.

More recently, there has been an emphasis on the possible effects of sound stimulations on the physiology and health of animals, suggesting the possibility of a ‘music therapy’ also for animals. The effects of music on animals can, however, largely vary according to the type of music played and most certainly according to the precise sound parameters concerned. Music is composed of frequencies (corresponding to the rhythm of repetitions: 10 Hz = 10 beats per second), frequency modulations and tempo which vary largely between genres of music.

However, many animal studies do not mention the type of music used, and still less its composition [2] and methodological issues may prevent definite conclusions about the impact of music, even on humans [8]. In humans, Calamassi and Pomponi [9]’s study shows that frequency parameters have an important effect on attention and mood, independent of the musical genre broadcasted. New Age and meditation musical compositions are generally expected to increase relaxation, which has been attributed by some authors to their tuning at A4 (LA3) 432 Hz (standard carrier frequency for instrument tuning until 1950) instead of the current official (since 1975) standard for tuning instruments of 440 Hz. The question therefore arises of whether different components could be involved in different mechanisms of action, e.g., melody with associative learning and frequency parameters with intrinsic (cellular, genetic) vibrational phenomena [10].

In the present study, we hypothesized that frequency parameters, more than music melody, would be of the highest importance in triggering changes in animals’ internal states. Therefore, we tested a commercial sound system based on the playback of an array of pure frequencies (i.e., clear of any melody effect or associative memory) at non-audible levels. The broad range of frequencies included EEG slow wave frequencies (theta-alpha) and pure-tone frequencies, all used in human music therapy and generated as multiples or dividers of the 432 Hz frequency. The aim here was not to test the effect of individual frequencies but to test the supposed optimal beneficial effect obtained through combining frequencies individually known for enhancing well-being. Horses are excellent domestic animal models to test for the efficiency of such sound stimuli, as they are sensitive to an array of frequencies through their hearing system, as well as other body parts [11,12]. This is because low frequencies are registered by horses not only through their sense of hearing but also in the form of ground vibrations, perceived by their hooves or even their teeth, for example, when they are grazing in a meadow [13]. The commonly restrictive conditions imposed on domestic horses lead to chronic welfare problems that are expressed through behavioral alterations such as abnormal repetitive behaviors, depression-like behaviors or aggressiveness, but also physiological (abnormal hematology or cortisol profile) and health (back problems, colic, etc.) disorders [14,15,16,17,18,19]. Racehorses are especially at risk, as they experience these restricted conditions of life (mostly single stall housing, high concentrate feeding), especially if they start intensive training at a very early age (2 years) [20,21]. Finally, horses’ resting EEG profiles are strongly influenced by their welfare state, with a higher prevalence of theta waves in animals experiencing good welfare and of gamma waves in animals in a compromised welfare state, especially if they have back problems [22,23]. Here, we tested the potential effect of daily exposure to a combination of selected frequencies, including theta and alpha EEG waves and frequencies used in human music therapy delivered via a specialized commercial audio frequency broadcast system, on the behavior and physiology of 12 racehorses in their home stalls. Many animal studies focus on immediate effects, such as respiratory rates, but here we tried to have an extensive view of possible chronic effects through an array of behavioral and physiological welfare measures. We followed an ABA crossover procedure for comparisons between playback/non-playback phases and looked for possible medium-term effects.

## 2. Materials and Methods

### 2.1. Population (See Also Appendix A)

Twenty racehorses, without any known health problem, were initially involved in the study, but because of changes (accidents requiring special care, sales, death on the racetrack, etc.), only 12 of them (10 Thoroughbreds and 2 French Chasers), 5 mares and 7 geldings, aged 3 to 8 years (4.4 +/−1.7), remained for the whole study p eriod (i.e., 9 weeks).

All horses were experienced with training and had been working in the facility for at least 1 month. They all lived in the same barn in one training facility (Mayenne, France) and thus had similar conditions of life.

The barn was organized into two rows that faced each other, separated by a walkway. The half doors of each stall opened on the walkway and there were metal bars enclosing the top half of the front of stalls. Horses were thus in a permanent single stall housing (12 m^2^ stall) that allowed visual, acoustic and very partial tactile contact with other horses. Stalls were bedded with sawdust and water was provided *ad libitum* through automatic drinkers. Daily feeding routine included hay twice a day (12 kg per day in total), once in the morning at 6:00 am and again in the evening at 4:00 pm, in racks placed on top of the side walls, thus above the horses’ heads, some additional minerals and 10.5 l of commercial racehorse pellets per day, distributed throughout the day (from 1:00 am to 10:00 pm) in seven meals through an automatic feeder. The horses were trained every morning (except Sunday) for 30 min at a nearby racecourse (10 min from the stable, at a slow trot). The intensity of the training was generally the same from one horse to the next (varying slightly according to the dates of each horse’s races). After training, the horses stayed in their stalls for the rest of the day. Each horse therefore had one day off a week (Sunday) and one more day if racing. Only 4 horses participated in at least one race in each phase. During the study, the horse management routine did not change (see Appendix A). There was no difference in time spent training (Friedman test, chi-square = NaN, df = 2, *p* = NA), nor in the number of races run (Friedman test, chi-square = 1.18, df = 2, *p* = 0.55) according to phase.

### 2.2. Study Phases and Playback Protocol

We followed an ABA crossover study design that is regularly used for such types of studies [7,24,25]. The study lasted 9 weeks, from 21 March to 24 May 2022, and was divided into three parts (Figure 1): a 3-week pre-playback phase (PRE), a 3-week playback phase (PL) and a 3-week post-playback phase (POST). Wieneska et al. [26] estimated that the sound stimulus they broadcasted daily had an effect on horses after 2–3 weeks; thus, a three-week exposure period was chosen.

The acoustic system used was a commercial system designed and sold by the company @Excelessens Inc. under the name ‘BioHarmonisation System’ and both the acoustic content of the stimulus and technical adaptations of the loudspeakers for vibrations’ propagation are under legal rights protection (Soleau n° 622092 250123 and 622093 250123). The funders had no role in the design of the study, in the collection, analyses or interpretation of data, in the writing of the manuscript or in the decision to publish the results. The authors played no role in the design of the signal or its propagation mode. In the interests of confidentiality, the authors were only informed of the characteristics of the signal and were not given the opportunity to verify its content.

Therefore, precise details about the choice of frequencies, their order of succession and the transmission of frequencies through air cannot be provided; only the general features are given here.

The main characteristics and principles of the stimulus were as follows:

The choice of frequencies was based on an optimization principle that consisted of adding frequencies known or supposed to have a positive effect on the mental or physical state: theta and alpha brain waves [27] and very low (LF) to very high (HF) frequencies used in human music therapy [28]. Thus, the range of frequencies was between 4 and 10,000 Hz. 86 different frequencies were presented as 2-s pure tones in a predetermined succession; thus, each sequence lasted 172 s (86 × 2 s) and was broadcasted again after the end of the previous one.

Since the session lasted 45 min, there were 15 repetitions of the sequence. All frequencies were multiples or dividers of the A4-432 Hz frequency, as it is supposed to promote well-being and attentiveness to the stimulus [9,29]. Frequencies were generated using a Siglent SDG1032X–Generator 30 MHz (SIGLENT Technologies Germany GmbH, Augsburg, Germany). The system is generally sold with additional meditation music added to the frequencies. In order to avoid possible additional effects of the added musical melodies and to test the effect of the frequencies per se, only the abovementioned pure frequencies were retained here for playback. Playback was performed through a HMA120 Ecler amplifier and three spherical Ecler loudspeakers (euC106) modified through an embedded module developed by @Excelessens (Soleau 622093 250123) that allowed a non-audible propagation of all experimental frequencies. The loudspeakers were distributed equidistant (9.5 m) in the 30-m long walkway of the barn, hanging from the roof well above the horses’ head (5 m high) (Figure 2). Horses were located at a distance of 2.8 ± 1 m from the closest loudspeaker, i.e., at homogeneous distances from the sound source.

All the equipment was set up on 14 March 2022 so as to leave time for habituation to the equipment and the playback started on 11 April in the evening and lasted until 3 May in the morning. During the playback phase, the broadcast took place (automatically) twice a day for 45 min: 10:00–10:45 p.m. and 5:00–5:45 a.m., so as to avoid the time periods when there was activity in the stable and any interference with possible effects of the stimulus on human caretakers/riders who arrived at 6:00 a.m. [30].

### 2.3. Assessment of Behavioral and Physiological States of Horses

We were particularly interested in assessing whether the stimulation could influence the behavior of horses as well as their internal state. Therefore, data were collected that included validated behavioral indicators of welfare (i.e., chronic states), emotional level (i.e., excitatory components) and time budget. Many studies on horses but also other farm animals have relied upon heart rate variables to assess emotional states. We chose not to use such a system as horses have associative memories of work or other ‘invasive’ human-related objects with past experiences [31,32] and this seems to be still more so the case for racehorses (Gueguen et al. in prep.). Moreover, heart rate is more involved in short-lived emotions, whereas the physiological changes observed reflected medium-term effects. Heart rate measures do not really help with durable states, as their correlation with stereotypic behaviors or pain is controversial [33,34,35] and changes in welfare states are not necessarily associated with heart rate changes [36]. Moreover, girthing is a common problem in racehorses due to possible health problems, and most of all, it would have been quite impractical.

Physiological data were based on hematological data, rather than measures of the HPA axis-related hormones, which are more ambiguous indicators of chronic stress [37], especially in horses [19,38,39]. Hematological measures are clearly influenced by horses’ welfare/chronic stress state [14,21,38,40] and especially training stress in racehorses [21,40].

Behavioral observations were performed by a single observer (MD) during the first and third weeks of each phase in each horse’s home stall. MD received extensive training for ethological observations until agreement reached 96% on a subset of data between LG and her (kappa coefficient of 0.85).

Horse training occurred every morning between 6:00 and 11:00 am, with a break at 8.00 am, so morning observations were performed during the break or late morning. In order to ensure having enough contexts to assess all possible types of behaviors [41,42], observation sessions were conducted during the three periods: pre-meal period (the 6:00 am hay distribution), in the morning (8:00 am and 11:00 am) and in the afternoon (14:15 pm and 15:50 pm). At that time of year, daylight occurred only after 6:00 am; thus, in order to avoid disturbing the horses during the dark periods, the observer wore an infrared lamp (Brand: Forclaz). The idea was to record the horses during the first pre-meal (hay) period. At the time of the first observations of the day (5:00 am), the stable was indeed dark until the caretakers arrived to feed the horses 30 min later at 5:30. The idea was to disturb as little as possible the horses’ routine. Voice recordings were made with a very low voice, and we did not observe behavioral changes when the experimenter recorded observations.

#### 2.3.1. Behavioral Measures

##### Welfare and Emotional Indicators (See Also Appendix B)

During 5-min sessions, the observer stood in the corridor in front of a horse’s stall and recorded (on a voice recorder (‘Olympus–digital voice recorder WS-852’) all occurrences of the following behaviors (Appendix B):-Welfare indicators: stereotypic behaviors (motor and oral) ([41]; see Appendix B and Appendix C). Many studies on horses focus on the ‘traditionally’ recognized stereotyped behaviors, but it is crucial to also take into account other more subtle behaviors to ensure a reliable assessment [17]. This is why, in addition to the ‘classical’ stereotypies, well known and described ([42], cribbing, windsucking, weaving), we have considered other abnormal repeated behaviors also cited and observed here (tongue playing, head movement, repetitive licking, etc. [15,43,44,45,46,47,48,49,50,51]).-Vacuum chewing (associated with lip licking) and yawning, which frequency increases in both chronic and acute stress situations [52]. Vacuum chewing is considered a stereotypic behavior in other species, such as cows or pigs [53].-Emotional indicators, including agitation behaviors (active walk, rears…) and frustration behaviors (pawing, vacuum threats, etc.) (see details in Appendix B and [54]).-Acoustic productions were also recorded as they can provide information on the internal state [55,56,57].

Individual horses were observed successively during the sessions, with a random order over the sessions. In total, we obtained five sessions (1 before feeding, 2 during the morning and 2 in the afternoon) per horse per week, leading to 10 sessions and thus 50 min per horse per phase.

##### Time Budget (Appendix D)

As in many ethological studies [44,48,58,59,60] the time spent by each horse in each activity was recorded using instantaneous scan sampling during direct observation sessions. There were many people working in these stables; therefore, there was nothing unusual about having a human walking around. The observer (MD) walked slowly (1 m/s) for 30 min along the walkway and recorded the horse’s behavior at the precise time she came across its stall starting randomly every time (e.g., 1st stall right-hand side and then left hand-side; 3rd stall left-hand side and then right-hand side, etc.), which corresponded to a scan every 2 min, i.e., 16 scans obtained for each [61]. The activities recorded (Appendix D) were feeding, resting (standing, sternal or lateral recumbency), observation (related to quiet states [62]), fixed gazes (related to possible apathetic states [63]) maintenance, locomotion, depressed posture, exploration (social and non-social) and repetitive/stereotypic behaviors. In total, we obtained five observation sessions per horse per week (i.e., 1 before feeding, 2 in the late morning and 2 in the afternoon). Each phase included two weeks of observations, so we performed 10 sessions (5 sessions × 2 weeks) of observation per horse per phase. This corresponds to 160 scans (16 scans per horse per session × 10 sessions) per horse and for each phase (PRE, PL, POST). The data were expressed as a percentage of scans for each activity.

#### 2.3.2. Physiological Measures

As a modification of their welfare state may impact the horses’ physiology [14,38], we investigated the possible relationships between the different phases and significant hematological parameters (complete blood cell count) of 11 of the 12 horses (one horse had to be excluded because it was ill at the time of the last sampling, i.e., at the end of the experiment).

Blood sampling is part of the routine sanitary control of racehorses in training facilities to monitor horses’ rehydration, muscular condition and food intake adequacy with exercise [64]. Therefore, the Rennes Ethical Committee for animal experimentation (CREEA) considered that no approval was required, since blood sampling was part of the usual examinations and was not conducted, especially for our experiment. It was undertaken with the approval of the facility’s manager and conducted by the usual experienced professional in charge of blood sampling on-site. All horses were used in the blood sampling procedure and accepted it well. It is worth noting that particular attention was given by the person in charge to minimize potential aversive effects of blood sampling, which was confirmed here by the absence of any reaction by the horses. The entire procedure, from halter fitting to the end of the sampling session, lasted less than 30 s.

Based on the facility routine, blood samples were collected before the horses left for training between 5:30 am and 11:00 am on 11 April (end of the pre-playback phase), 3 May (end of the playback phase) and 24 May (end of the post-playback phase and of the experiment). In addition to routine sampling, five milliliters of blood were collected in heparinized EDTA K3 tubes (BD Vacutainer^®^, Becton, Dickinson and Company, Franklin Lakes, NJ, USA), then placed in a refrigerator at 4 °C before being sent to a veterinary laboratory for analysis (LABEO Franck Duncombe, Caen, France) within the day.

Hematological data analyzed were the number of red blood cells (RBC), white blood cells (WBC), platelets (Pl), hemoglobin (Hb), mean corpuscular hemoglobin concentration (MCHC), mean corpuscular volume (MCV), and percentages of each type of leucocyte (neutrophils, eosinophils, basophils, lymphocytes, monocytes).

The standards of the laboratory that carried out the analyses are not those for racehorses, so the standards of another laboratory (Rossdale Laboratory [65]) specializing in thoroughbreds were used.

### 2.4. Statistical Analyses

Data consisted of actual individual numbers for the physiological data, percentage of scans per activity per horse and number of behavioral occurrences over 10 all occurrence sessions (50 min). As the data were not normally distributed (Shapiro-Wilk test, *p* < 0.05), non-parametric statistical tests were used. The statistical tests were performed using R 4.2.0 software (alpha threshold set at *p* < 0.05).

Friedman and subsequent Wilcoxon tests were used to evaluate the behavioral and physiological changes in relation to the experimental phase, i.e., between pre-playback, playback and post-playback phases. A Benjamini–Hochberg (BH) correction was applied for multiple comparisons [66].

Only statistically significant results are mentioned in the results section. The whole set of results can be found in Appendix E.

## 3. Results (See Also Appendix E)

### 3.1. Pre-Playback Phase

Before playback, welfare indicators revealed a ‘classical’ situation for single stall housing (e.g., 11 out of 12 horses exhibited at least one stereotypic behavior (Mean ± standard deviation, X ± ES = 10.5 ± 14.8 occurrences in 50 min), 7 a depressed posture (3.8 ± 5.2), and all expressed agitation or frustration behaviors (46.6 ± 21.6). Stereotypic behaviors were principally oral (4 horses, 6.8 ±12.4) and some motor (2 horses, 3.7 ± 7.1). Five horses expressed both (details of the number of horses performing each type of behavior in Appendix E). On the more positive side, all but 1 horse produced at least one snort during observations (3.4 ± 2.2).

The time budget was distributed between feeding (27.0 ± 9.8%) and resting (19.0 ± 9.9%), followed by observation behavior (16.0 ± 10%), time spent in fixed stances (15.0 ± 10%) or stereotypic behaviors (13.0 ± 10%).

Hematological data revealed that all horses were within reference standards for hemoglobin (Hb) and eosinophils. A few horses were, however, outside norms for some parameters: five were slightly below the standard for red blood cells (RBC) (standard: 9–14 × 10^6^/mm^3^, horses outside norms: mean ± standard deviation of horses, X ± ES = 8.2 × 10^6^ ± 6.1 × 10^5^), two were below the standard for Mean corpuscular hemoglobin concentration (MCHC) (standard: 32–36 g/dL, horses outside norms: X ± ES = 32 ± 0.007), five for platelets (standard: 127–206/mm^3^, horses outside norms: X ± ES = 104 ± 26) and three for monocytes (standard: 2–6%, horses outside norms: X ± ES = 1 ± 0.7), while two were above the norm for neutrophils (standard: 36–68%, horses outside norms: X ± ES = 70 ± 2.5) and one was above the norm for white blood cells (WBC) (standard: 4–10 × 10^3^/mm^3^, horse outside norms: 12 × 10^3^/mm^3^). Mean cell volume (MCV) and lymphocyte counts were above the norms for all horses (MCV: 48.0 ± 4.0; norm: 41–41%; lymphocytes: 22.0 ± 4.3; norm: 1.4–4.7fl). There were significant behavioral and physiological changes between the study phases. Only parameters for which there were statistically significant results are reported here (complete statistics in Appendix E).

### 3.2. Behavioral Changes According to the Study Phase (Figure 3)

There was a statistically significant reduction in the number of stereotypic behaviors (*p* = 0.03), particularly between the PRE and POST phases (Wilcoxon V = 73, BH correction: *p* = 0.02), but none between the PL/PRE and PL/POST phases (PL/PRE: *p* = 0.12; PL/POST: *p* = 0.12) (Figure 3a). Horses showed fewer agitation behaviors and vacuum chewing during the POST phase than during the PRE phase (agitation: *p* = 0.03; PRE/POST: *p* = 0.04; vacuum chewing: *p* = 0.006; PRE/POST: *p* = 0.04) (Figure 3b,c).

**Figure 3 animals-13-02970-f003:**
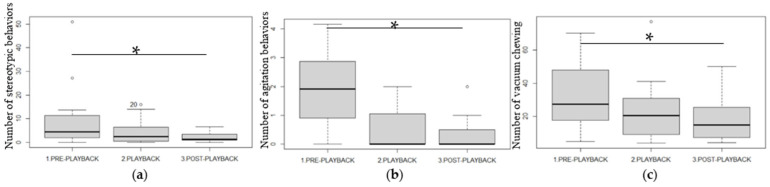
Number of (**a**) stereotypic behaviors, (**b**) agitation behaviors and (**c**) vacuum chewing in 50 min of all occurrences observation (median for all horses) during the three phases of the study. Friedman’s and Wilcoxon (with BH correction)’s tests: * *p* < 0.05. The plots show median (line within box), 25th and 75th percentiles (box), 5th and 95th percentiles (whiskers), and outliers (degree sign).

### 3.3. Changes in Time Budget (Table 1, Figure 4)

Several behavioral differences were noted between phases (Table 1). All the other results are shown in the table in Appendix E.

**Figure 4 animals-13-02970-f004:**
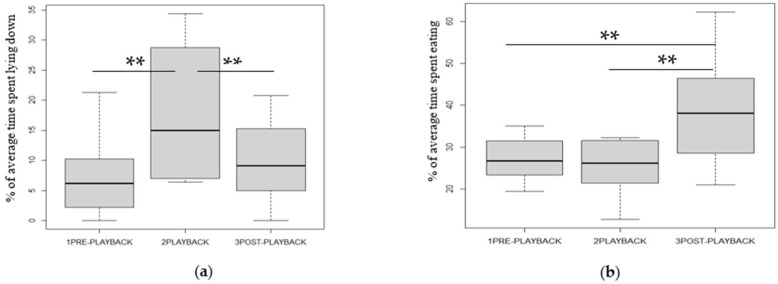
Percentage of median population time spent: (**a**) in recumbency (sternal and lateral), (**b**) eating hay. Friedman’s and Wilcoxon (with BH correction)’s tests: ** *p* < 0.01. The plots show median (line within box), 25th and 75th percentiles (box), 5th and 95th percentiles (whiskers), and outliers (degree sign).

**Table 1 animals-13-02970-t001:** Statistics on significant behavioral differences according to the study phases.

Behavior	Mean, Standard Deviation, % of Minimum and Maximum Time for Each Behaviour for Each Period			Friedman Test, df = 2	Comparisons of Phases 2 to 2: Wilcoxon with BH Correction		
	PRE	PL	POST		PRE/PL	PL/POST	PRE/POST
Sternal recumbency	5.9 ± 5.2	13.8 ± 7.4	7.5 ± 5.4	*p* = 0.0004	*p* = 0.002	*p* = 0.002	
	Min 0	Min 4	Min 0				
	Max 17.4	Max 25	Max 17				
Lateral recumbency	0.9 ± 1.2	4.0 ± 4.2	2.7 ± 2.7	*p* = 0.02	*p* = 0.05		
	Min 0	Min 0	Min 0				
	Max 3.9	Max 12	Max 9				
Observation	15.6 ± 6.4	16.4 ± 9.3	11.4 ± 5.7	*p* = 0.02			
	Min 6	Min 8	Min 5				
	Max 29	Max 34	Max 22				
Hay feeding	27.1 ± 5.3	25.5 ± 6.1	38.7 ± 11.9	*p* = 0.002		*p* = 0.003	*p* = 0.004
	Min 20	Min 13	Min 21				
	Max 35	Max 32	Max 62				
Exploration of environment	2.7 ± 1.8	1.0 ± 1.2	1.6 ± 1.8	*p* = 0.03	*p* = 0.01		
	Min 1	Min 8	Min 5				
	Max 7	Max 34	Max 21				
Fixed gazes	15.2 ± 7.7	5.7 ± 3.9	5.0 ± 3.0	*p* = 0.0001	*p* = 0.0007		*p* = 0.0007
	Min 7	Min 4	Min 5				
	Max 36	Max 34	Max 22				

There were differences in the time budgets according to the study phase. The horses showed quieter states after the playback started. They spent more time from the broadcast on in recumbent resting, whether sternal or lateral. They also spent more time, from diffusion onwards, in quiet observation of their environment and eating hay. On the contrary, they spent less time exploring (mostly sniffing) their physical environment or conspecifics and in fixed gazes.

### 3.4. Physiological Changes According to the Study Phase (Figure 5)

There were physiological changes according to the study phase.

**Figure 5 animals-13-02970-f005:**
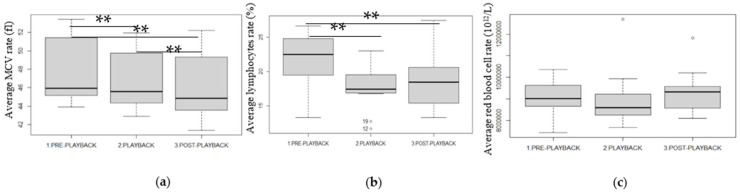
Examples of hematological data over the study period. (**a**) Average MCV rate; (**b**) average lymphocyte rate; (**c**) average red blood cell rate of the population during the three phases of the study. Friedman’s and Wilcoxon (with BH correction)’s tests: ** *p* < 0.01. The plots show median (line within box), 25th and 75th percentiles (box), 5th and 95th percentiles (whiskers), and outliers (degree sign).

Two parameters increased: MCHC (*p* = 0.0003; PRE/PL: *p* = 0.003; PRE/POST: *p* = 0.009), and neutrophils (*p* = 0.0008; PRE/PL: *p* = 0.003; PRE/POST: *p* = 0.009), but finally all horses were within the norms for both parameters in the post-playback phase. There was some decrease in white blood cells, but levels remained within norms (*p* = 0.03). There was a decrease in the lymphocyte counts during the playback phase (only) (*p* = 0.002; PRE/PL: *p* = 0.003; PRE/POST: *p* = 0.01) and for the mean corpuscular volume (MCV), which kept decreasing during and still after the playback (*p* = 0.00009; PRE/PL: *p* = 0.002; PRE/POST: *p* = 0.002; PL/POST: *p* = 0.01). These decreases were not large enough to have levels within norms at any stage, but in the post-playback phase, for the MCV, 8 out of the 12 horses were close to norms (between 41 and 44) against 3 in the pre-playback phase. Furthermore, no difference was found in red blood cell and hemoglobin rates.

## 4. Discussion

This study, where 12 racehorses were submitted twice daily to the playback of an array of pure tone frequencies non-audible to the humans in the environment) during three weeks (i.e., a total of 21 h of observation) in their home stalls, is, to our knowledge, unique both in the system used and in the extent of the observed major effects on both behavior and physiology.

Overall, a number of behavioral changes were found during the broadcast phase, which cannot be attributed to a change in routine. No environmental factors differed between the phases, which could explain the changes observed after the start of the broadcast. Horses exhibited less stereotypic behaviors and vacuum chewing, less agitation behaviors and fixed gazes, but more recumbent resting (both sternal and lateral), more quiet observation behaviors and hay feeding during the exposure to the frequencies. The MCHC, a red blood cell-related parameter, increased, lymphocytes decreased and most of all the MCV, which was well above norms for all horses in the pre-playback phase, decreased steadily over time, so that most horses were close to norms at the end of the experiment.

A few of these results were observed only during the playback phase (increase in lateral recumbency), but in most cases, they were still observed in the three weeks following the cessation of playback (e.g., decrease of agitation behaviors, vacuum chewing, fixed gazes, MCV). Some effects even still increased or appeared significant only after the playback had ceased, such as the increase in hay ingestion, in neutrophils, and the decrease in MCV. Overall, horses appeared quieter and in better welfare states after the playback was initiated and even after it stopped, revealing at least a medium-term durability of the treatment. Finding ways of diminishing the stress associated with restricted conditions of life and training for racing, including facilitating physical recovery, is an important challenge for the welfare of racehorses, and possibly racing performance (Stachurska et al. [67] found positive effects of music on prize money won by horses).

Effect of Acoustic Stimulation on Horses’ Behavior and Physiology

Few studies have been performed on the possible effects of acoustic stimulations on horses and all used music, which is a multivariable stimulation (i.e., music is made up of different frequencies, frequency modulations and tempo, which vary considerably from one musical genre to another). In her pioneering study on 9 ponies, Houpt et al. [68] suggested that country music had a calming effect. Hartman and Greening [24], who investigated the behavior of 7 horses during nights before, during and after periods of two hours daily playback of Beethoven Symphonia n°9, reported an increase in feeding behavior and recumbency during the playback phase. Using the same music, Huo et al. [25] reported an increase of feeding behavior in their four experimental horses during the playback period as compared to before and after (3 days periods). In a study performed on 70 Arabian racehorses [31] receiving the broadcast of animal-designed music (J. Marlow New Age music, a specialist in the music for animals) through loudspeakers also designed for animals (My Pet Speaker, Pet Acoustics Inc., passing band: 200–12,000 Hz, https://www.petacoustics.com/, accessed on 13 June 2023), the authors mention first effects after a month. However, these effects diminished after 2–3 months of playback, suggesting that the maximum duration of a similar stimulation should not exceed two months. Stachurska et al. [67] also found that the experimental horses had a better performance at racing (e.g., increased sum of gains). A similar experiment performed on geriatric horses using the same system also led to calmer states after 2–3 weeks of broadcast but with no lasting effect [26]. Differences in protocols prevent direct comparisons with our results, but increased feeding and calmer states appear to be common effects observed at least during playback of classical, country or New Age music and our pure-frequencies stimulus. Most of the previous horse studies used longer daily playback durations and the last study was also on much longer periods [26]. The duration of our sessions (2 × 45 min daily) was determined by the technical service of Excellessens on the basis of pre-tests performed on other animals. Different studies also indicate that three weeks is a reasonable duration for observing music-related changes in internal states [26]; consequently, we used this duration, and indeed strong effects were already visible within this 3 week period. However, given the results obtained here, and especially the physiological results, a longer period may have been still more beneficial.

The hematological data for the horses in our study group were overall within normal ranges for racehorses and although there were some statistically significant changes in values over the study period, they were mostly associated with a return to normal values.

One parameter, the MCV, had values that were initially abnormally high for all horses compared to the published norm for racehorses. These values steadily decreased after the playback started and remained lower after the playback ceased [65]. It is therefore remarkable that these values clearly and steadily decreased after the playback started and remained lower after the playback ceased. Training stress is associated, especially for young horses, with increased levels of MCV [69] and overall, high values of red blood cell-related parameters reflect a fatigue state [70]. We cannot know what mechanism could explain this outcome, but it is quite possible that recuperation after training improved after exposure to the frequencies as they spent more time lying down and were less agitated during playback and the phase afterwards. Furthermore, it is known that deep sleep is achieved mainly when horses are in lateral recumbency [71], and that sleep quality is essential in sports for physical recovery [72]. It is also possible that the horses in the present study improved their quality of sleep over the course of the study. In addition, stereotypic horses have been found to lie down less than non-stereotypic horses [33,49].

We did not assess performance given that only a few horses were involved in a limited number of races over the course of the study; thus, further investigation would be required to identify any potential positive effects. The results obtained in the abovementioned studies and the present one tend to fit with those reported in other domestic and captive species, for which the playback of slow rhythm music, classical or ‘New Age’, seems to be associated with increased calm ([2] for a rev. for farm animals, dogs: [73,74,75]). Similar positive effects such as decreased agitation and increased relaxation (a general term covering measures such as increased respiration rate, decreased heart rate or quieter behaviors such as resting) have also been described in a variety of species including domestic dogs, cattle, pigs and chicks, as well as captive birds, elephants and gorillas [2,73,74,75,76,77,78], although results can be contradictory and preferences for silence over music can be observed in some species (marmosets: [79]; orang-outans: [80]). Indeed, playback of music has been associated with increased growth rate in different species of fish, lower respiration rate in different mammals, increased milk yield in cows or relaxation in pregnant sows (review in [2,81,82,83]). However, few studies went beyond measuring heart rate and respiration rate variables in terms of physiological measures, but the playback of ‘Vivaldi Four Seasons’ music to chicken has been associated with lower corticosterone and increased weight gain [84]. Fast rhythm music of rock or metal types seems to invariably induce avoidance and opposite effects to slow rhythm music in humans and animals, ranging from decreased blood pressure in humans, metabolic problems in cattle, stress reactions in pigs to weight loss in chickens [79,84,85,86]. Cattle respond physiologically to different types of music [81] and milk yield may increase by up to 12.6% for Indian music [87]. More intriguing still are the findings of Papoutsoglou et al. [83] and Çatlı et al. [88] showing increased growth in carps and turbot fish, respectively, when hearing Mozart sonates, and still more those showing that even deaf breams may grow better when submitted to such music [82]. In laboratory male rats, playback of ‘Mozart Sonata’ K.448 may attenuate anxiety- and depression-like behaviors and increase hippocampal spine density, i.e., cognition/memory [89]. This same music had an anxiolytic effect on female rats as compared to silence, while white noise had an anxiogenic effect [90]. Playback of the ‘Vivaldi Four Seasons’ composition to zebrafishes has led to reduced anxiety and differences in gene expression (decreased proinflammatory cytokines) [91]. The playback of Mozart’s String quartet was associated with stress reduction, a lower heterophil to lymphocyte ratio and a more symmetrical development of wings and legs in chicks [92].

Few studies have gone as far as looking at the neurophysiological mechanisms that might explain these results and they have focused solely on music and not on specific frequencies. For example, Banerjee et al. [93] recorded EEG signals from 10 humans under three experimental conditions: rest, with music and without music and EEG recordings of alpha, theta and gamma brain waves. It was found that the alpha frequency bands were strengthened when listening to music and that the EEG profiles changed according to the type of music. Changes in theta waves were also observed between pleasant and unpleasant musical extracts [84]. Changes in brainwaves linked to the audition of music could be associated with behavioral changes, particularly for persons suffering from depressive disorders [87].

What sound?

We are clearly in the preliminary stages of understanding what can explain the observed positive or negative effects of sound stimulation. Interestingly, the same ‘Vivaldi Four Seasons’ music induced relaxation and positive states in pregnant sows but more agitation in piglets [86,94,95,96,97], which raises in particular the question of how animals perceive human-created music. Snowdon et al. [98] have suggested that music for animals should be closer to species-specific types of sounds and showed that cats were more attracted to “cat-music” (i.e., music composed of sounds inspired by cats’ species-specific sounds). However, humans also liked the “cat-music”. Commercial systems have been developed, some designed specifically for domestic animals, such as the New Age ‘J. Marlow music’ generally broadcasted through specifically designed playback systems (MyPet Speaker, Pet acoustics Inc.), broadly used in recent studies [26,67]. However, in animals as well as humans, there is a general lack of knowledge about the possible mechanisms underlying the supposed or demonstrated effects of sound stimulation. Clements-Cortes and Bartel [10] have proposed an explanatory model with four levels for humans: (1) learned cognitive process (e.g., associative memories); (2) cognitive activation of neural circuits (e.g., sensorimotor training); (3) stimulated neural coherence (neuronal synchronization through specific frequencies, rhythmic sensory stimulation); (4) cellular and genetic levels: neural cells in vitro with nerve growth factor exposed to sounds between 10 to 20,000 Hz during 30 min show an increased neurite growth for 10–1000 Hz [99]. A sound may become meaningful and induce a particular emotional valence through experience, even in animals such as horses [54] and piglets [100], but this does not need to be musical. Different rhythms do not have the same effect on turbot fish growth [88]. Further studies should aim to investigate the effects of the components of music on the physical and mental health of a large range of species (including humans). Such comparative studies would be key for identifying shared basic processes.

Overall, when given the choice, animals prefer slow rhythms and classical or country music over rock and metal genres [7]. Playback of classical music was associated with a decreased number of stereotypic behaviors, less agitation, and more resting in cows, while rock and metal music had opposite effects in both cows and pigs [85,86]. Particular music types such as the above-mentioned ‘Mozart Sonata’ K.448 or K.525, ‘Vivaldi Four Seasons’ or Beethoven Symphonia n °9 have been particularly used in human and animal studies (‘Mozart effect‘: [94,95,96]), but why and how these music types may have an effect on mental and/or physical state remains poorly understood.

Obviously, this goes far beyond analogies with species-specific sounds [101]. How can we explain the convergence of effects in a broad range of species on specific music genres? How can we explain that this works with deaf species, and still more so with yeast cells or in vitro neural cells [99,102]. Clements-Cortes and Bartel [10] propose that the future relies upon understanding better the third and fourth levels of their model (see above): synchronizing neural activity and actions at the cell and gene levels.

The system we tested here allowed us to get rid of the possible effects of melody and auditory ‘conscious hearing’. Although we do not know the precise mechanism involved, playback of these pure tones was associated with changes in the horses’ behavior and physiology.

Although the precise choice and order of frequencies were determined by the technical service of Excellessens and remained unknown for us, the effects were clear and we had some elements that can help future research. The results support our hypothesis that melody may not be the key to understanding the beneficial or detrimental effects of some music genres. The frequencies involved may be one major, if not sufficient, key. Active audition may not be necessary: non-audible environmental noise is a major public health problem [103], and as noted previously, deaf fish and humans react to sounds, as do mere cells.

Similar to other horse studies, we found an increase in ingestion behaviors after or during the broadcast period. Russo et al. [104] found an increase in the secretion of ghrelin, associated with eating behavior, in the hypothalamus of rats exposed to music. There are synergic interactions between Ghre and dopamine in the hypothalamus that influence feeding motivation. The broadcast of acoustic frequencies between 4000 Hz and 16 kHz promotes dopamine synthesis and reduces blood pressure [105]. Dopamine enhances optimism in humans [106], while dopamine depletion is believed to be part of the factors involved in the development of stereotypic behaviors and learning deficits in horses [107] as well as other domestic animals [103]. Such physiological effects and their consequences can require more time to develop and last longer than other changes, which would explain that changes in these behaviors (increased eating and less stereotypic behaviors) were more visible after the playback phase in our study. Unfortunately, most evidence comes from human or laboratory animals and to our knowledge, no study on domestic or captive animals has investigated whether the carrier frequency had an impact. New Age/meditation studies are mostly tuned on the A4-432 Hz and some authors have attributed these effects to their tuning instead of the current official (since 1975) standard for tuning instruments of 440 Hz. Different musicians (including Verdi) and opera singers claimed that 432 Hz tuning enhanced audibility, whereas 440 Hz enhanced agitation and aggressiveness [9,108]. Di Nasso et al. [29] found that the anxiety of patients in endodontic (dental care) treatment decreased if they heard a music tuned at 432 Hz. This view was reinforced by Menziletoglu et al. [109]’s findings, who, based on different measures such as a visual analogue scale, questionnaires or salivary cortisol measures, reported that anxiety was lower when the music broadcasted was tuned at 432 Hz rather than at 440 Hz. In a more experimental type of setting, Calamassi and Pomponi [9] broadcast a variety of music genres tuned at both 432 and 440 Hz to participants who were blind to the tuning. The aim of this study was to identify differences in participants’ vital parameters (e.g., heart and breathing rates), perceptions (e.g., fatigue and stress), levels of concentration (assessed by nurses using a specific observation grid on different criteria) and general satisfaction (assessed using a questionnaire) with the experience of listening to this different frequency music. They found that the same participants showed greater concentration during the broadcast and expressed greater satisfaction with the music tuned to 432 Hz. While listening to music tuned to 440 Hz, participants were more agitated, more critical of other participants and less satisfied. According to the authors, since 432 Hz is a multiple of the Schumann Earth resonance (8 Hz) described by NASA, it would be in harmonic alignment, whereas 440 Hz would be in harmonic misalignment. To our knowledge, these different tunings have not been explicitly tested in animals, but some species show preferences for consonant music (i.e., the perception of consonance and dissonance is an enigmatic aspect of this issue). Whereas consonance is static and evokes a pleasant feeling, dissonance is dynamic and intense, and gives rise to an unpleasant feeling [101,110,111,112,113]. As mentioned above, the commercial animal-designed music used in some studies is based on New Age music, likely to be tuned at 432 Hz. Similarly, meditation music is often tuned to 432 Hz and studies suggest that it induces relaxation states in humans by increasing the proportion of slow EEG waves, i.e., theta (4–8 Hz) and alpha (10–14 Hz) brain waves in resting EEG profiles [114,115,116,117]. For decades now, it has been hypothesized that the ‘sonification’ of brain activity and its playback could influence the subject’s internal state (a ‘music of the brain’: [118,119]). Neurofeedback, where human participants learn to increase these particular waves, has become an important therapeutic tool as it has been shown to help reduce symptoms of several neurological and psychiatric disorders [120]. Similarly, laboratory cats can be trained to use this interactive playback and thus resist substance-induced epileptic seizures [120]. Neurofeedback using theta waves (4–7Hz) has been associated with reduced alcoholism [121]. More recently still, tools have been developed to produce ‘musical compositions’ based on the extraction of particular frequencies from the global EEG profile [118].

What mechanisms of sound transmission and its effects?

Horses perceive low-frequency sounds through the ears but also hooves and teeth at least [26]. Sound consists of the propagation of vibrations in the air. The ciliary cells in the cochlea transform these sound vibrations into electrical impulses that travel via the auditory nerve to the brain (primary auditory cortex). However, as mentioned above, other body parts and cells can be involved in the detection of air-borne vibrations. Vibrations caused by sounds not only reach the auditory apparatus but also resonate, leading to possible tactile perception through the mechanoreceptors of the somatosensory system [122]. The accuracy of this sensory modality for sounds appears to be enhanced in deaf humans [123]. Moreover, there is evidence of the influence of sound vibrations in vertebrate and non-vertebrate species that lack an auditory system (gilthead sea bream: [83]; snails: [124]) and in vitro neuronal cells [99]. Aggio et al. [102] not only showed that sounds had an influence on yeast cell metabolism (i.e., increased growth rate and reduced biomass production), but metabolic and physiological responses varied with the type of sound that was played (i.e., high or low frequency sonic vibrations). It has been suggested that specific frequencies may not only affect hearing but also physical functions [125] and that there are different means of propagation of vibrations [126], which explains the nociceptive effects of non-auditory environmental noise [127]. According to Muehsam and Ventura [128], one hears sounds not only through the ears but rather through the whole body, cells being detectors of mechanical vibrations. Very low frequencies (100 Hz) have also been shown to impact physiological or physical aspects such as balance in young adult humans [125] and frequencies between 27 and 113 Hz broadcasted to elderly patients have been associated with a reduction of bone loss [129].

In conclusion, the present stimulus involved an ensemble of (a) low frequencies (<100 Hz) (known for improving balance in young human adults [119]); (b) high frequencies; (c) slow EEG waves (used in neurofeedback to enhance relaxation [120]); and (d) multiples/dividers of the 432 Hz tuning frequency. It may well be thus that the stimulus used here was indeed an optimization of these known frequencies’ effects, as the racehorses’ behavioral and physiological changes observed show that they were sensitive to this array of selected non-auditory pure tone frequencies.

It also seems that evolution may have shaped acoustic mechanisms, resulting in convergences between taxa in the perception and integration of acoustic information (see also [130]). Additional research is required on current studies that have reported the emotional and physiological effects of exposure to certain famous musical compositions. A greater understanding of how not only the auditory system but also the whole body processes the information is needed. In the present case, the system was shown to be a potential tool to assist the racehorses in dealing with both the welfare problems related to their conditions of life (single stall housing, high concentrate diet) and the regular intense stress of training and racing. Like any ‘enrichment’ tool, it will not replace the need for re-thinking practices in training facilities and favors, such as providing all horses with access to social companions [131], free movement [17], more balanced diets [15] and maybe less stressful training, leaving space for times of relaxation.

### Limitations

The sample size was fairly small. Future studies could be conducted on larger samples of horses at different facilities and under different conditions and where a multivariable analysis can be conducted to adjust for potential confounders. We could not use the whole initial group of 20 horses because of events that were rather common in training facilities, and we needed to have all horses in similar conditions in order to limit the impact of other management factors. The ABA crossover protocol has proved useful in a variety of studies, but it would also have been good to have control horses without playback over the same time period. However, given the large propagation distance provided by the system, we could not be sure that horses in neighboring barns of the same facility would not be affected to some extent by the frequencies broadcast. All the more so as sound could be perceived by horses not only through the ears, but also through the whole body, the cells being detectors of mechanical vibrations. Finally, using a commercial system allowed us to test the hypothesis that pure frequencies, selected based on known effects on humans, could be effective even though they were non-audible and not associated with melodies. However, we cannot at this stage demonstrate which of them (e.g., EEG frequencies? low frequencies?) were more determinant, whether all of them were needed or whether their order of succession was optimal.

Furthermore, we have only indirect evidence of what they perceived (through the changes observed during the broadcast period, which was not attributed to a change in routines). An increase in recumbency behaviors may reflect better recuperation via increased sleep time. However, in our observations, eye opening (one of the criteria for REM sleep) was not always visible. So, even if the time spent lying down is very interesting, we can’t say that this means the horses spent more time asleep, even if this remains a serious hypothesis (REM sleep occurs mainly during lying down).

Duration of playback was standard for this commercial system, but it would be interesting to test different durations of both playback bouts and overall procedures. In particular, changes in physiological parameters may be slower than for behaviors and require longer periods of playback: one can wonder whether a longer playback period could have ensured that all horses were in the norms for MCV, for example. Finally, it would have been interesting to look at performance, but unfortunately, a few horses (only 4/12 horses raced over the course of the study) went out for only a few races over the course of the study.

## 5. Conclusions

The results of this study (1) show that, before the sounds were broadcasted, the observed racehorses showed compromised welfare, with most of them exhibiting stereotypic behaviors, agitation and/or high frequencies of yawning and vacuum chewing, as well as abnormally high levels of lymphocytes and MCV (considered as an indicator of training stress); (2) that there was an improvement on all these parameters during and for most of them, also or still more after the cessation of the broadcast period. These findings open new lines for future research involving studies with larger population sizes and control horses, as well as examination of the effects of individual frequencies. This would promote improved welfare, which, in turn, may lead to improved performance. It would also help us better understand how non-audible stimuli might have the observed effects on their mental and physical health.

These studies should also lead to more awareness of welfare issues and their consequences in the racing professional world and the importance of taking measures to improve it. Broadcasting stimuli may alleviate some stress effects, but management decisions also have to be revised thoroughly in order to improve the living and working conditions of racehorses.

## Figures and Tables

**Figure 1 animals-13-02970-f001:**
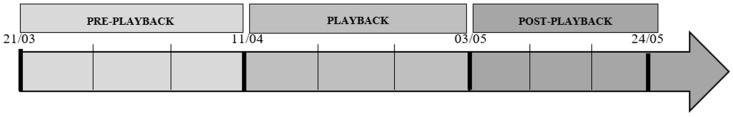
Temporal organization of the study.

**Figure 2 animals-13-02970-f002:**
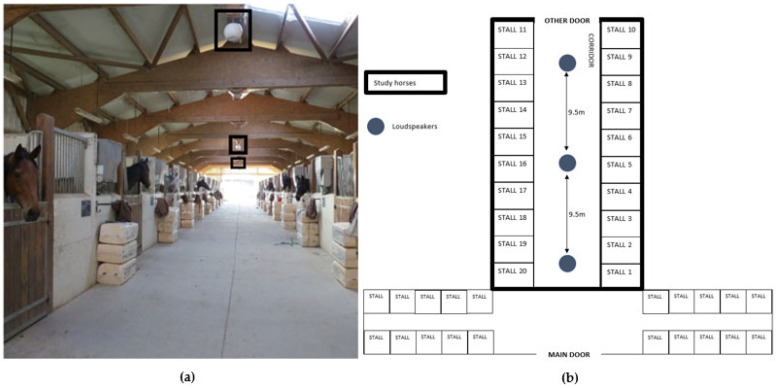
(**a**) Loudspeakers and their distribution in the barn. (**b**) Map of the barn with loudspeakers’ positions.

## Data Availability

Data are available on reasonable request to the corresponding author.

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
