# Peer review of "Selected Acoustic Frequencies Have a Positive Impact on Behavioural and Physiological Welfare Indicators in Thoroughbred Racehorses"

_animals, 2023, doi:10.3390/ani13182970_

Round 1

Reviewer 1 Report

Review: animals-2307710

This paper describes a study in which thoroughbred racehorses were exposed to an inaudible standardised sound source and their response monitored before, during and after a three week period of exposure. The authors' stated intention was to determine whether the exposure had an impact on physiologic and behavioural parameters, and by extension, implications for welfare. The authors have tried to be thorough in their approach and have presented the topic and their findings with palpable enthusiasm. The paper is thoughtfully laid out and extensively reviews relevant literature. The topic is quite novel and of great interest and could well have a significant positive impact on racehorse management, if only by virtue of the awareness it could create. The authors are to be congratulated, and will hopefully continue their studies!

I have concerns, however, with regard to funding source and declaration, some of the methodology and experimental design, and with the authors' enthusiasm and subsequent overreach with regard to interpretation of findings. In the interests of scientific objectivity, these issues should be addressed before this very interesting piece of work is placed before the Journal's readership. My comments address general, then specific issues.

"Excelessens" appears to be a commercial enterprise, and is not sufficiently clearly declared as such in the paper. It is referenced as a "society", when the correct translation would be "company" or similar. The fact that the funding source was a commercial enterprise whose equipment was being profiled should be much more clearly identified, since otherwise, despite the authors' declaration, the readers' suspicions are reasonably aroused concerning the authors' objectivity. This is especially so when findings are presented with such enthusiasm and with relatively limited consideration of other possible explanations for findings. Also, while acknowledging the manufacturer's reluctance to disclose the proprietary details of their product, the limited information the authors have been able to provide concerning the nature of the signal they broadcast significantly limits the scientific value of the study's findings and further erodes confidence in the paper's objectivity.

The paper is long, and while thorough, especially the introduction, and bearing in mind methodological and interpretive issues addressed below, the authors might consider presenting the general material as a review of the topic. This could be a shorter paper introducing the need for investigation. The actual study can then be presented as a preliminary investigation in a separate paper, with findings and interpretations being conditional on the need for further investigation, as referenced by the authors. Otherwise, in the interests of brevity and succinctness, much of the otherwise very interesting review material would need to be removed to make the paper more manageable.

While acknowledging the significant difficulties involved in gathering data in a working racing stable, the sampling of time budget data is suspect. Such data are usually gathered from video recordings over a much longer time period. Scan sampling in 30 minute epochs scarcely seems sufficient, while the novelty of the observer's appearance in the stable greatly increases the possibility that their presence influenced behaviour. Additionally, this effect would have been greater at the start than the end of the sampling period. These issues do not appear to have been given sufficient consideration by the authors when interpreting and commenting on the significance of their findings or when referencing limitations.

A change of title is required. The study was not able to determine specifically what was being played, while the relevance to welfare is interpretive and secondary, lying at least one level beyond the parameters measured. A suggestion would be "A Preliminary Study of Behavioural and Physiologic Responses of Thoroughbred Racehorses to Standardised Background Sounds and Implications for Welfare". As it stands at present, the title, like much of the text, tends to over-interpret findings.

The study does not pay sufficient attention to other plausible explanations for findings, particularly in an experimental setup in which the authors could neither control the environment nor the selection of subjects. This is perhaps the greatest weakness of the study and deserves greater attention, particularly when it comes to assessing the significance of findings. For example, what was the age distribution of subjects, how long had they been in the stable and what was their stage of training, what was the daily training regimen, how were work days off work handled in terms of data collection and the weight attached to observations, what medications were subjects receiving. In other words, did any routine management considerations influence study findings, was study design adequately considerate of such influences?

The researchers avoided turning on lights so as not to disturb the horses during quiet times, yet used a voice recorder to record observations. What was the reason for this approach, convenience, was the stable very dark? What was the relationship between the time the observer entered the stable and the timing of routine procedures such as feeding? How were such issues controlled - early morning is a particularly problematic time in a racing stable?

If the sound was 100% inaudible, how do you know the horses were actually sensing anything, consciously or subconsciously? This needs explicitly addressing - are you assuming that sounds that lie below the auditory threshold (amplitude and frequency are both of relevance here) nonetheless have an impact? And is this threshold considerate of species - Is the auditory threshold that of the horse or that of humans? Passing reference is made to ways in which sound vibrations can be detected (through the feet, for example), but the authors do not deal directly with how they suspect the sounds played could influence the horses, even though they touch on the issue in their literature review.

The study has no external controls by which events outside the study group could be assessed. For example, a group managed in the same way as the experimental subjects, subject to the same regimen and isolated from the sound signals, yet subject to all other environmental and management influences. Admittedly, for this to work you would need several barns so as to control for a barn effect.

Line 345 - apologies, but I cannot understand how you arrive at 160 scans per horse, and what exactly is a period - pre-, during and post? A table would help here.

Line 450 - this paragraph is hugely complicated by the listing of statistics and short forms, so much so as to be almost unintelligible. While acknowledging the technical correctness of the structure, it might be better to reference a table so as to make the findings clear, and to eliminate components such as degrees of freedom.

Figure 4 - what is the significance of the fact that time spent lying down was greatest during the playback period?

Line 500 - so were they within norms r above norms?

Line 505 - "mean" corpuscular volume

Use of English is generally very good and most issues are minor and can be dealt with in copyediting. There are one or two problems that are recurrent…

Line 87 and elsewhere - "Four Seasons"

Line 630 and elsewhere - "Mozart Sonata"

Figure 5C - please clarify. Image says "average red blood cell rate", legend says "red blood cell levels", y-axis implies RBC count, the text in the paragraph above refers to MCMC (what is this, do you mean MCHC - line 494) and does not reference red blood cell numbers or haemoglobin. In all of figure 5 y-axis labels refer to "rate" when the reference should be to the correct name of the parameter that is being measured together with the units of measurement. Paragraph 3.4 and figure 5 need serious revision.

Paragraph 3.4, line 492, this paragraph and the discussion needs to give greater consideration to factors that might have influenced these parameters other than the sounds played. For example, the horses appear to have been raced quite infrequently during the study period - what else was happening to them in this period? Might the horses have been dehydrated initially? The changes you describe in red cell parameters (assuming that MCMC means MCHC) are suggestive of resolving red cell loss. Comparing your results from your own laboratory with those of a different laboratory, even one dedicated to assessing racehorses, is somewhat problematic - does your own laboratory not have its own normal values? In many racing stables the track PA system plays in the shedrows and stables, and some horses can be noted to have quite significant responses to some sounds such as the bugles often played before a race. Were you able to control all other sound and vibration sources?

Line 537 - what does "+20 on the lateral sides" mean? Paragraph 3.5 is very difficult to interpret. 31 horses involved in falls compared with 10+6+1 = 17 - so the rest were in the non-test barns? Why did you not compare the two barn areas, and how exposed were the other horses to the sound? The fall rate in the 12 horses appears to be 1.42 per horse for the period compared with 0.7 for the remainder of the barn. The rate was particularly high in the first period of the study - it is unsafe to refer to this as an effect of the sound treatment when such events are sufficiently infrequent to be highly subject to chance events. Alternatively, one might construe that the study barn contains a relatively high number of problem horses, in which case you might conclude that the sound effect was much greater! A much more circumspect conclusion might be drawn.

The fact that the specific frequencies and sounds that were included in the package were unknown to the authors is problematic, and perhaps more so than suggested in the paper. Indeed, since the sounds were not audible the authors cannot provide any evidence that the subjects were actually exposed to sounds. It would not have been difficult to monitor the output of the devices and even to analyse any signal obtained, though it is acknowledged that this might have been contrary to the arrangements made with the supplier. Nonetheless, this might actually be the ultimate in use of the "black box" whereby the researchers and the readers are required to have faith that what is described as having been done was actually carried out.

It would be helpful, since the paper already goes into so much detail in reviewing the literature, if the authors differentiated between carrier frequency and acoustic frequency. Carrier frequency describes the central frequency used in signal transmission between source (for example a radio mast) and the receiver (for example, a radio receiver set). Acoustic frequency represents the sound frequencies into which the modulation information embedded in the carrier signal is encoded for delivery to a receiver where it is decoded for delivery, for audio signals, to a speaker or equivalent. To the extent that the average clinician considers frequencies, those frequencies are audio frequencies and cover a much broader range than the very narrow carrier frequency, which may be arcane to some.

Appendix C.

Degrees of freedom are described in the header row - they do not need to be repeated in each cell.

What is MCMV?

Please declare in the table what units are being used - it should be possible to interpret a table without any reference to the text.

Author Response

Reviewer 1

The authors have tried to be thorough in their approach and have presented the topic and their findings with palpable enthusiasm. The paper is thoughtfully laid out and extensively reviews relevant literature. The topic is quite novel and of great interest and could well have a significant positive impact on racehorse management, if only by virtue of the awareness it could create. The authors are to be congratulated, and will hopefully continue their studies!

Thank you for this positive statement

I have concerns, however, with regard to funding source and declaration, some of the methodology and experimental design, and with the authors' enthusiasm and subsequent over reach with regard to interpretation of findings. In the interests of scientific objectivity, these issues should be addressed before this very interesting piece of work is placed before the Journal's readership.

The funding source had been declared quite clearly in the devoted part of the paper and we also clearly stated that the funders had not interfered in the design of the study nor the interpretation of results. Since it was obviously not clear enough, we repeat this statement now in the methods part (lines 278-282).

We were ourselves quite surprised by the extent of the changes in the welfare measures during the broadcast period, which may explain our enthusiasm which in any case does not overreach the results (see our further responses).

"Excelessens" appears to be a commercial enterprise, and is not sufficiently clearly declared as such in the paper. It is referenced as a "society", when the correct translation would be "company" or similar. The fact that the funding source was a commercial enterprise whose equipment was being profiled should be much more clearly identified, since otherwise, despite the authors' declaration, the readers' suspicions are reasonably aroused concerning the authors' objectivity. This is especially so when findings are presented with such enthusiasm and with relatively limited consideration of other possible explanations for findings

Actually « société » is the French name for a company, so we made here a translation mistake, sorry for this misunderstanding. We had the feeling that the commercial aspect was well mentioned in the methods part already, but we have added a paragraph to insist on it (lines 278-282) (see above).

We have been some hat surprised by the reviewers’ suspicions about our interpretations of the results. None of the authors has any commercial link with the funders and if any every study funded by a private (or public) company was suspicious, a large part of the literature on horses could be considered as such. We develop further why the other explanations for the results proposed by the reviewer, which of course had been examined, are not tenable. But we agree that these remarks were useful precisely for testing alternate hypotheses and thank the reviewer for this critical view.

The paper is long, and while thorough, especially the introduction, and bearing in mind methodological and interpretive issues addressed below, the authors might consider presenting the general material as a review of the topic. This could be a shorter paper introducing the need for investigation. The actual study can then be presented as a preliminary investigation in a separate paper, with findings and interpretations being conditional on the need for further investigation, as referenced by the authors. Otherwise, in the interests of brevity and succinctness, much of the otherwise very interesting review material would need to be removed to make the paper more manageable.

Making a separate review was a good idea, and we did consider it.  However, it still needed to be associated with the experimental part. After discussion with Animals’ board, it appeared that it would be costly and difficult to have both papers in this issue. Therefore, and since the Reviewer 2 did not require it, we decided to leave the introduction in this paper, especially as it helps answering some of the reviewer’s concerns as well (see further). 

While acknowledging the significant difficulties involved in gathering data in a working racing stable, the sampling of time budget data is suspect. Such data are usually gathered from video recordings over a much longer time period. Scan sampling in 30-minute epochs scarcely seems sufficient, while the novelty of the observer's appearance in the stable greatly increases the possibility that their presence influenced behaviour. Additionally, this effect would have been greater at the start than the end of the sampling period. These issues do not appear to have been given sufficient consideration by the authors when interpreting and commenting on the significance of their findings or when referencing limitations.

We have been very surprised by this comment as a large majority of ethological studies on horses has been made by direct observations (e.g., Mactaggart & Phillips, 2023 ; KW et al, 2023 ; Lundqvist & Mülle, 2022 ; Waters et al, 2002 ; Ransom et al, 2010).  There were many people working in these stables and walking around, therefore there was nothing unusual about having a human walking around. A paragraph and bibliographical references have been added to the text on this point (line 382-385). The behavioural sampling used was quite classical (see also e.g : Ransom et al, 2010 ; Bulens et al, 2013 ; Ninomiya et al, 2007 ; Christensen et al, 2022 ; Hanis et al, 2020) and led to a « classically » large amount of data. Obviously, our descriptions were not clear enough and there has been a misunderstanding as the reviewer did not understand how we reached 160 scans per horse (see comment below), it is made clearer (lines 393-398) now which should respond to both comments and reassure the reviewer we hope.

Apologies, but I cannot understand how you arrive at 160 scans per horse, and what exactly is a period - pre-, during and post? A table would help here.

Obviously, our descriptions were not clear enough and there has been a misunderstanding as the reviewer did not understand how we reached 160 scans per horse. We have therefore added the following paragraph “In total, we obtained five observation sessions per horse per week (i.e. 1 before feeding, 2 in the late morning and 2 in the afternoon). Each period included two weeks of observations, so we performed 10 sessions (5 sessions x 2 weeks) of observation per horse per period. This corresponds to 160 scans (16 scans per horse per session x 10 sessions) per horse and for each period (PRE, PL, POST). The data were expressed as a percentage of scans for each activity.” (lines 393-398).

A change of title is required. The study was not able to determine specifically what was being played, while the relevance to welfare is interpretive and secondary, lying at least one level beyond the parameters measured. A suggestion would be "A Preliminary Study of Behavioural and Physiologic Responses of Thoroughbred Racehorses to Standardised Background Sounds and Implications for Welfare". As it stands at present, the title, like much of the text, tends to over-interpret findings.

Despite no request from Reviewer 2 on this aspect, we changed the title to “A first study on the positive impact of selected acoustic frequencies on behavioural and physiological welfare indicators in thoroughbred racehorses”. However, we want to keep the notion of frequencies as these were pure tones which makes a big difference with the frequency modulations found in music. We did measure validated welfare indicators, so we are not sure what the reviewer means by “relevance to welfare is interpre(ta)tive”.

The study does not pay sufficient attention to other plausible explanations for findings, particularly in an experimental setup in which the authors could neither control the environment nor the selection of subjects. This is perhaps the greatest weakness of the study and deserves greater attention, particularly when it comes to assessing the significance of findings. For example, what was the age distribution of subjects, how long had they been in the stable and what was their stage of training, what was the daily training regimen, how were work days off work handled in terms of data collection and the weight attached to observations, what medications were subjects receiving. In other words, did any routine management considerations influence study findings, was study design adequately considerate of such influences?

We did of course consider all possible explanations before writing the paper but we agree that we did not list them and thus this was not clear enough in the manuscript as it stood. We have now added information about age, medications, time spent in the stable, training stages and number of races. Daily training regimen and handling of days off had already been described but we have expanded on this (lines 243-248 and Appendix A).

As the reader will thus see now, there was no change in routine management that could explain such collective changes in behaviour and physiology between phases.

The researchers avoided turning on lights so as not to disturb the horses during quiet times, yet used a voice recorder to record observations. What was the reason for this approach, convenience, was the stable very dark? What was the relationship between the time the observer entered the stable and the timing of routine procedures such as feeding? How were such issues controlled - early morning is a particularly problematic time in a racing stable?

We explain now more extensively these methodological choices (lines 350-357). At that time of year, daylight occurred only after 6:00 am thus, in order to avoid disturbing the horses during the dark periods, the observer wore an infrared lamp (Brand: Forclaz). The idea was to record the horses’behaviour during the first pre-meal (hay) period. At the time of the first observations of the day (5.00 am), the stable was indeed dark until the caretakers arrived to feed the horses 30 min. later at 5.30. The idea was to disturb as little as possible the horses’ routine. Voice recordings were made with a very low voice and we did not observe behavioural changes when the experimenter recorded observations.

If the sound was 100% inaudible, how do you know the horses were actually sensing anything, consciously or subconsciously? This needs explicitly addressing - are you assuming that sounds that lie below the auditory threshold (amplitude and frequency are both of relevance here) nonetheless have an impact? And is this threshold considerate of species - Is the auditory threshold that of the horse or that of humans? Passing reference is made to ways in which sound vibrations can be detected (through the feet, for example), but the authors do not deal directly with how they suspect the sounds played could influence the horses, even though they touch on the issue in their literature review.

As mentioned in the limitations part (line 794-801), we have only indirect evidence that they perceived it (through the drastic changes observed during the broadcast period and which could not be attributed to any change in the routines). We have now added elements on possible transmission mechanisms, which were described in the literature (see the detailed introduction part, line 191-196) and we insisted that this is a first study that needs to be confirmed by further investigations as stated now: “This first study would clearly deserve to be repeated on larger samples of horses and different facilities within and outside racing.” (Line 779-780).

The study has no external controls by which events outside the study group could be assessed. For example, a group managed in the same way as the experimental subjects, subject to the same regimen and isolated from the sound signals, yet subject to all other environmental and management influences. Admittedly, for this to work you would need several barns so as to control for a barn effect.

We agree that this would be a good addendum for further validation in future studies and had already mentioned it in the limitations part (lines 783-787). This is also why we agree that it is better to indicate “a first study” in the title. The ABA procedure is however “classical” and has been validated in an array of studies (see refs in the manuscript, lines 260-261).

Line 345 - apologies, but I cannot understand how you arrive at 160 scans per horse, and what exactly is a period - pre-, during and post? A table would help here.

This was obviously unclear, we have rewritten this part as there were obviously confusions between « pre-feeding » times and pre-broadcast phases. See lines 393-399. We think we are also solving other issues at the same time (see comment above).

Line 450 - this paragraph is hugely complicated by the listing of statistics and short forms, so much so as to be almost unintelligible. While acknowledging the technical correctness of the structure, it might be better to reference a table so as to make the findings clear, and to eliminate components such as degrees of freedom.

We agree and have now put all the stats into a dedicated table (Table3). Thanks for the suggestion

Figure 4 - what is the significance of the fact that time spent lying down was greatest during the playback period?

REM sleep has been observed in horses, particularly when lying down, associated with rapid eye movements and rhythmic ear contractions (Williams et al, 2008). During our observations, we couldn't see the horses' eyes, so it's possible that they were in the sleep phase. Furthermore, it has been shown that internal (e.g. drugs) and external (e.g. sound) factors also influence the dynamic process of sleep (Halasz, 1998). Continuous auditory stimulation during the night (e.g. music) can have a masking and relaxing effect in animals (Hartman & Greening, 2019 ; Wells & Irwin, 2008). For example, in horses, music during the night appears to facilitate the onset of biologically significant behaviors, including lateral decubitus, with behaviors maintained beyond the enrichment period (Hartman & Greening, 2019).

It is also important to note that the recumbent posture can be linked to the animal's state of welfare. For example, it has been found that stereotyped horses lie down less compared with non-stereotyped horses (Clegg et al, 2008 ; Hausberger et al, 2008).

Finally, sleep is an important biological need for all living mammals, due to its restorative properties and cognitive role in memory consolidation (Greening et McBride, 2022).

A sentence has therefore been added on this subject in the results section: “This may reflect a longer period of sleep, since REM sleep is most common in horses during the lying-down phase.”  Line 508-510

A paragraph has also been added to the discussion section: “We cannot know what mechanism could explain this outcome, but it is quite possible that recuperation after training improved after exposure to the frequencies, as they spent more time lying down and were less agitated during playback and the period afterwards. Furthermore, it is known that deep sleep is achieved mainly when horses are in the lateral recumbency (Williams et al, 2008), and that sleep quality is essential in sport for physical recovery [109]. In addition, continuous auditory stimulation during the night (e.g. music) can have a masking and relaxing effect in animals (Hartman & Greening, 2019; Wells & Irwin, 2008). For example, in horses, music at night appears to facilitate the onset of biologically significant behaviors, including lateral decubitus, with behaviors maintained beyond the enrichment period (Hartman & Greening, 2019). It is therefore also possible that the horses in the study improved their sleep quality over the course of the study. In addition, stereotyped horses have been found to lie less compared with non-stereotyped horses (126, 127). This posture could therefore be linked to indicators of welfare and reflect the animal's internal state, as well as enabling better physical recovery.” Line 657-669

A sentence was also added in the limits about the fact that we couldn't see the horses' eyes when they were lying down, to affirm that they were in the sleep phase : “The increase in recumbency behaviors is interesting, and may reflect better recuperation via increased sleep time (Greening and McBride, 2022). However, in our observations, eye opening (one of the criteria for REM sleep, Williams et al, 2008) was not always visible. So, even if the time spent lying down is very interesting, we can't say that this means the horses spent more time asleep, even if this remains a serious hypothesis (REM sleep occurs mainly during lying down, Greening and McBride, 2022).” Line 798-801

Line 500 - so were they within norms or above norms?

The sentence has been rewritten: “but finally all horses were within the norms for both parameters in the post-playback phase” (line 538-539)

Line 505 - "mean" corpuscular volume

Line 87 and elsewhere - "Four Seasons"

Line 630 and elsewhere - "Mozart Sonata"

Thank you, all these points have been changed throughout the manuscript.

Figure 5C - please clarify. Image says "average red blood cell rate", legend says "red blood cell levels", y-axis implies RBC count, the text in the paragraph above refers to MCMC (what is this, do you mean MCHC - line 494) and does not reference red blood cell numbers or haemoglobin. In all of figure 5 y-axis labels refer to "rate" when the reference should be to the correct name of the parameter that is being measured together with the units of measurement. Paragraph 3.4 and figure 5 need serious revision.

MCMC was a typing error, it is MCHC.

We have therefore reviewed figure 5 (revised the legend and axis titles with units). Part 3.4 has also been revised, adding the missing information on red blood cells and hemoglobin (“Furthermore, no difference was found for red blood cell and haemoglobin rates.” line…) and standardizing the information with figure 5.

Paragraph 3.4, line 492, this paragraph and the discussion needs to give greater consideration to factors that might have influenced these parameters other than the sounds played. For example, the horses appear to have been raced quite infrequently during the study period - what else was happening o them in this period? Might the horses have been dehydrated initially? The changes you describe in red cell parameters (assuming that MCMC means MCHC) are suggestive of resolving red cell loss. Comparing your results from your own laboratory with those of a different laboratory, even one dedicated to assessing racehorses, is somewhat problematic -does your own laboratory not have its own normal values? In many racing stables the track PA system plays in the she drows and stables, and some horses can be noted to have quite significant responses to some sounds such as the bugles often played before a race. Were you able to control all other sound and vibration sources?

Many of these questions are now answered in the results part, as we have added all available information on horses’ characteristics and environmental conditions along the different parts of the study (training, races, experience, medications… (see above and appendix A). None of these factors differed between phases and therefore can explain the changes observed after the broadcast had started. Actually, these data show that there is indeed variability, as in any field study, but the fact is that there was the same amount of variability (in age, number of races per horse, medications given etc…) throughout the study and quite independently of the broadcast status. Moreover we do not see why the bugles played before a race would lead to have behavioural or physiological changes specifically during our stimulus broadcast phase?

Anyway, we are grateful to the reviewer for raising these concerns as these additional informations are certainly preventing future readers to wonder about their possible influence. More precisely, the horses raced equally infrequently during the three phases (pre-, during and post- broadcast). The changes in MCHC have been discussed in view of the current literature and there is no clear reason they would be dehydrated and still less to think this would have changed then as routine procedures did not change … In brief there is nothing that can suggest that changes in routines or the environment were exactly tuned to the experimental phases to the point of explaining the results. We have also added now a paragraph in the discussion on these aspects: “Overall, a number of behavioural changes were found, during the broadcast phase, which cannot be attributed to a change in routine. In fact, no environmental factors differed between the phases which could explain the changes observed after the start of the broadcast.” (lines 585-588)

Line 537 - what does "+20 on the lateral sides" mean? Paragraph 3.5 is very difficult to interpret. 31 horses involved in falls compared with 10+6+1 = 17 - so the rest were in the non-test barns? Why did you not compare the two barn areas, and how exposed were the other horses to the sound? The fall rate in the 12 horses appears to be 1.42 per horse for the period compared with 0.7 for the remainder of the barn. The rate was particularly high in the first period of the study - it is unsafe to refer to this as an effect of the sound treatment when such events are sufficiently infrequent to be highly subject to chance events. Alternatively, one might construe that the study barn contains a relatively high number of problem horses, in which case you might conclude that the sound effect was much greater!

Thank you for your comments. The fall rate was under 1 per phase per horse and the horses we had added were at the end of the stable, thus within hearing range but further up. We agree that the rarity of this phenomenon prevents definite conclusions and therefore we have removed these results and all references to them from the manuscript.

It would not have been difficult to monitor the output of the devices and even to analyse any signal obtained, though it is acknowledged that this might have been contrary to the arrangements made with the supplier. Nonetheless, this might actually be the ultimate in use of the "black box" whereby the researchers and the readers are required to have faith that what is described as having been done was actually carried out.

We agree on this point, but as mentioned by the reviewer in his/her first statement, the idea was first of all to raise awareness about the feasibility and interest of using these types of frequencies. Other studies have used the “Pet speaker” system which does not provide its precise contents either to users [31].

We have added a point on this in the materials and methods section:  « The authors played no role in the design of the signal or its propagation mode. The authors were only informed of the main characteristics of the signal, and were not given the opportunity (in the interests of confidentiality) to verify its content. lines (281-282).

It would be helpful, since the paper already goes into so much detail in reviewing the literature, if the authors differentiated between carrier frequency and acoustic frequency. Carrier frequency describes the central frequency used in signal transmission between source (for example a radio mast) and the receiver (for example, a radio receiver set). Acoustic frequency represents the sound frequencies into which the modulation information embedded in the carrier signal is encoded for delivery to a receiver where it is decoded for delivery, for audio signals, to a speaker or equivalent. To the extent that the average clinician considers frequencies, those frequencies are audio frequencies and cover a much broader range than the very narrow carrier frequency, which may be arcane to some

As said above, the authors were only informed of the main characteristics of the signal, and did not have the possibility (for reasons of confidentiality) to verify its content. It is therefore unfortunately not possible for us to differentiate between carrier and acoustic frequency, although we agree that it would have been very useful.

Appendix C.

Degrees of freedom are described in the header row - they do not need to be repeated in each cell.

What is MCMV?

Please declare in the table what units are being used - it should be possible to interpret a table without any reference to the text.

All these suggestions have been considered. MCMV was a typing error, this was changed for MCHC. line (534).

Reviewer 2 Report

This paper "Playback of selected acoustic frequencies is associated with in-2 creased welfare in thoroughbred racehorses" is a very interesting work in a novel area that based on the results reported here, has promise to improve the welfare of stabled racehorses.  The paper is suitable for publication with amendments.

The literature review in the introduction is thorough and canvases a wide range of relevant sources.  However, critical details about some references/ideas are lacking in places.

The method is reasonably well described however some details are missing and this is noted in the annotated document.

The discussion and conclusions are mostly soundly drawn from the data presented in the results however some components are overly speculative and should be more cautiously presented.  There is insufficient attention paid to the possible mechanisms of action of the frequencies in relation to the observed effects.  While I appreciate this area of study is in its infancy, a greater attempt to explore putative physiological/neurological explanations would have assisted to more closely align the existing literature with the findings of this study.

I am concerned about how equine stereotypy behaviours are defined in this paper as it appears to me that a new definition is being used that is not in alignment with the extensive existing literature on equine stereotypies.   I have made extensive notes on this issue in the annotated copy of the paper and this should be addressed prior to publication as the results rest heavily on changes in behaviour following exposure to frequencies.  Consequently, it should be very clear to the reader what the specific behaviours were that changed.

I have made extensive notes in regards to improving the clarity of the writing because it is not as clear as it could be.  In some cases I have made suggestions as to possible ways to reword some sentences-these are a suggestion only.  In other areas the writing requires more significant amendment to improve the clarity and I have left it up to the authors to determine the most appropriate changes.

The written presentation of some of the results is confusing and hard to follow and requires major restructuring prior to publication.  The graphs and figures are very clear and complement the text very well.

The reference list is extensive and relevant.  Some additional references are required as per the annotated copy of the paper.

In summary, this paper makes an interesting and important contribution to the literature and explores a promising method for potentially improving the welfare of stabled racehorses and likely stabled horses in general.  The paper is suitable for publication subject to the amendments I have identified.

Author Response

Reviewer 2

This paper "Playback of selected acoustic frequencies is associated with increased welfare in thoroughbred racehorses" is a very interesting work in a novel area that based on the results reported here, has promise to improve the welfare of stabled racehorses. The paper is suitable for publication with amendments.

Thank you very much for this positive statement

The literature review in the introduction is thorough and can vases a wide range of relevant sources. However, critical details about some references/ideas are lacking in places.

Thank you for your suggestions in the annotated MS, we have taken them all into account

A number of changes have been made in the introduction as requested

Here, we tested the potential effect of daily exposure to a combination of selected frequencies, including theta and alpha EEG waves and frequencies used in human music therapy delivery via a specialised commercial audio frequency broadcast system. What is important is the frequencies, so it should be mentioned at the start of the sentence and the means by which the frequencies were delivered should be second.

This paragraph: “Here, we tested the possible impact of the daily playback of a commercial system based on a combination of selected frequencies, including theta and alpha EEG waves and frequencies used in human music therapy, on the behavior and physiology of 12 racehorses in their home stalls” has been changed to “Here, we tested the potential effect of daily exposure to a combination of selected frequencies, including theta and alpha EEG waves and frequencies used in human music therapy delivery via a specialised commercial audio frequency broadcast system on the behavior and physiology of 12 racehorses in their home stalls” line 214-217

Change the sentence to : The half doors of each stalls opened

Line 93: Define cat music.

Added: i.e. music composed of sounds inspired by cats’ species-specific sounds (now line 93-94)

Line 107: Give example of some species to which this applies to provide context/relevance for horses.

Addition of two references: 35, 36 (line 108)

We changed it for: further studies should aim at investigating the effects of the components of music on the physical and mental health of a large range of species (including humans). Such comparative studies would be a key for identifying shared basic processes. (now lines 109-111)

The method is reasonably well described however some details are missing and this is noted in the annotated document.

We have taken into account all your comments

Line 341: these need to be clearly defined/described.

The table in Appendix C has been added to provide an overview and detailed description of stereotypical behaviours.

The discussion and conclusions are mostly soundly drawn from the data presented in the results however some components are overly speculative and should be more cautiously presented. There is insufficient attention paid to the possible mechanisms of action of the frequencies in relation to the observed effects. While I appreciate this area of study is in its infancy, a greater attempt to explore putative physiological/neurological explanations would have assisted to more closely align the existing literature with the findings of this study.

In response also to reviewer 1, we have increased the limitations part and changed the title. Studies have focused on music in general rather than on specific acoustic frequencies. Nevertheless, it has been shown that music can induce changes in activity in fundamental brain structures up to and including behavioural changes (Ramirez et al, 2015; Banerjee et al, 2016; Liu et al, 2020; Trochidis & Biguand, 2012). This is discussed in more detail in the discussion section with this new paragraph: “Few studies have looked at the neurophysiological mechanisms that might explain these results. However, the few studies that have done so have focused solely on music and not on specific frequencies. For example, Banerjee et al (2016) recorded EEG signals from 10 humans under three experimental conditions: rest, with music and without music and a frequency analysis was performed for alpha, theta and gamma brain rhythms. It was found that the alpha frequency bands were strengthened when listening to music. In addition, the frequency of the waves varied according to the type of music. This is particularly true of the difference in the frequency of theta waves observed between pleasant and unpleasant musical extracts (Trochidis and Biguand, 2012). Going further, these changes in brainwaves linked to the broadcasting of music can have a direct impact on behaviour, particularly in people suffering from depressive disorders. The latter show a significant decrease in relative alpha activity in their left frontal lobe, which can be interpreted as a reduction in their depressive disorder (Ramirez et al, 2015).”  Line 693-705

Multivariable stimulation: Not clear what is meant by this - please specify. 

Addition of an explanation: Multivariable stimulation (i.e. music is made up of different frequencies, frequency modulations and tempo, which vary considerably from one musical genre to another). Line 613-615

I am concerned about how equine stereotypy behaviours are defined in this paper as it appears to me that a new definition is being used that is not in alignment with the extensive existing literature on equine stereotypies. I have made extensive notes on this issue in the annotated copy of the paper and this should be addressed prior to publication as the results rest heavily on changes in behaviour following exposure to frequencies. Consequently, it should be very clear to the reader what the specific behaviours were that changed.

In order to respond to Reviewer 2 and be clearer for readers, we have added the following praragraph: “The recent existing literature has very much moved from restricting the term of stereotypic behaviours to the “traditional” stereotypies towards including the many different forms of repetitive behaviours horses do perform under inappropriate conditions. Indeed, abnormal repetitive behaviours, whether called as such or as stereotypies reflect poor welfare and recent studies have shown the importance of considering all of them (ref?). For example,  repetitive stall circling /pacing; clapping of lips, tongue movements outside the mouth (“tongue play”), tongue rolling, head bobbing, nodding , repetitive licking or chewing of substrate or objects, vacuum chewing, door banging, head tossing and pawing have been defined as stereotypic behaviours by authors (Borthwick et al, 2022, Mactaggart & Phillips, 2023 ; Budzynska et al, 2022, Ruet et al. 2022, Pearson et al, 2022, KW et al. 2023, Mactaggart & Phillips 2023 , Dai et al, 2023, Lesimple et al. 2016, Lesimple 2020, Hausberger et al. 2009 amongst others). A previous study has shown that the same horses could change from one type of abnormal repetitive behavior to another, including “classical stereotypies” when housing conditions changed (Lesimple et al. 2019). Finally, in other species, such as cattle or pigs, tongue play, vacuum chewing or repetitive chewing of substrate are considered as stereotypies (e.g. Mattiello et al, 2005; Binev, 2022; Vandresen, 2022). For all these reasons, we have pooled here the “classical” stereotypies with the “abnormal repetitive behaviours” under the same term of stereotypic behaviours”. (lines 363-369). Testing the changes during the study according to the types of abnormal repetitive behavior performed appeared very difficult given the low number of horses performing each type (see appendix C).

What is the "system constructor" here?  Please explain what is meant by this and also why did this person/thing choose the duration?

We have replaced this sentence by : “The duration of our sessions (2 x 45‘ daily) was determined by the technical service of Excellessens on the basis of pre-tests performed on other animals .” (line 640-641)

Who was the producer?  It is still not clear.  The role of this person, persons needs to be clearly explained in the methods section.

We have added information: Although the precise choice and order of frequencies were determined by the technical service of Excellessens and remained unknown for us, the effects were clear and we had some elements that can help future research. Line 725-727

Unclear: “Such physiological effects and their consequences can take time to install and last longer than other changes, which would explain that changes on these behaviors (increased eating and less stereotypic behaviors) were more visible after the playback period in our study”

Thanks for this proposition that we applied:  Such physiological effects and their consequences can require more time to develop and last longer than other changes, which would explain that changes on these behaviors (increased eating and less stereotypic behaviors) were more visible after the playback period in our study. Line 741-744

This paragraph: The stimulus was involved low frequencies (<100 Hz) known for improving balance in young human adults [53], high frequencies and multiples/dividers of the 432Hz tuning frequencies, the former A4 carrier frequency for tuning instruments that has been replaced by the 440 Hz frequency worldwide officially since 1975 that renowned musicians have claimed as being best for hearing and well-being [e.g. 40]” has been changed to “The stimulus involved an ensemble of (a) low frequencies (<100 Hz) (known for improving balance in young human adults [53]); (b) high frequencies;  (c) slow EEG waves (used in neurofeedback to enhance relaxation [54]); and (d) multiples/dividers of the 432Hz tuning frequency. The 432Hz frequency is the former A4 carrier frequency for tuning instruments that has been replaced by the 440 Hz frequency worldwide officially since 1975. Renowned musicians have claimed the 432 Hz was being best for hearing and well-being [e.g. 40]” line 755-761

Additional research is required than current studies which have reported the emotional and physiological effects of exposure to the works of certain famous composers.  A greater understanding of not only ...

This paragraph: “Research has to go further than observing that some famous music pieces are inducing emotional and physiological changes and try to develop a better understanding on how not only the auditory system but also the whole body processes the information.” has been changed to “Additional research is required to current studies which have reported the emotional and physiological effects of exposure to the works of certain famous composers. A greater understanding of not only the auditory system but also on how the whole body processes the information is needed” line 767-770

Was shown to be a useful tool to assist the racehorses ...

This paragraph: “In the present case, such systems obviously can be useful tools for helping racehorses to deal….” has been changed to “In the present case, the system was shown to be a useful tool to assist the racehorses to deal with…” line 770-776

Such as providing all horses with access to ...

Overall, horses exhibited less.... during the exposure to the frequencies.

This paragraph: “Thus, overall, after the playback started, the horses exhibited less stereotypic behaviors and vacuum chewing, less agitation behaviors and fixed gazes, but more recumbent resting (both sternal and lateral), more quiet observation behaviors and hay feeding.“ has been changed to “Horses exhibited less stereotypic behaviors and vacuum chewing, less agitation behaviors and fixed gazes, but more recumbent resting (both sternal and lateral), more quiet observation behaviors and hay feeding during and to some extent still after  the exposure to the frequencies” line 588-591

Please provide a reference for this as the data in your study does not address this issue.

What were the differences?  Did they go up/down as a result of the exposure to the difference frequencies.  You are making an argument about the beneficial effect of the frequencies on ingestion behaviour, so you need to provide the detail of that effect from the reference you are citing.

Improved after exposure to the frequencies, as they spent more time lying down and were less agitated during playback and the period aftewards.

This paragraph “We cannot know what mechanism could explain this outcome, but it is quite possible that recuperation after training was better, since horses, while in their stall after training, layed down more and were less agitated during the playback and even post playback phase. “ has been changed to “We cannot know what mechanism could explain this outcome, but it is quite possible that recuperation after training improved after exposure to the frequencies, as they spent more time lying down and were less agitated during playback and the period afterwards. Furthermore, it is known that deep sleep is achieved mainly when horses are in the lateral recumbency (Williams et al, 2008), and that sleep quality is essential in sport for physical recovery [109]. In addition, continuous auditory stimulation during the night (e.g. music) can have a masking and relaxing effect in animals (Hartman & Greening, 2019; Wells & Irwin, 2008). For example, in horses, music at night appears to facilitate the onset of biologically significant behaviors, including lateral decubitus, with behaviors maintained beyond the enrichment period (Hartman & Greening, 2019). It is therefore also possible that the horses in the study improved their sleep quality over the course of the study. In addition, stereotyped horses have been found to lie less compared with non-stereotyped horses (126, 127). This posture could therefore be linked to indicators of welfare and reflect the animal's internal state, as well as enabling better physical recovery.” Line 657-669

Please check this.  Excess dopamine release in response to rewards is associated with the development of stereotypy, not decreased dopamine. Decreased dopamine is associated with learning deficits because dopamine is essential for synaptic plasticity particularly for operant learning and the development of habits.  However the literature on the effect of stereotypy and equine cognition is mixed depending on the type of cognitive task tested. 

This paragraph “Dopamine enhances optimism in humans [115] while dopamine depletion is believed to be part of the factors involved in the development of stereotypic behaviors in horses [116] and other domestic animals [117].” has been changed to “Dopamine enhances optimism in humans [115] while dopamine depletion is believed to be part of the factors involved in the development of learning deficits in horses [116] and other domestic animals [117].” Line 739-744

I have made extensive notes in regards to improving the clarity of the writing because it is not as clear as it could be. In some cases I have made suggestions as to possible ways to reword some sentences-these are a suggestion only. In other areas the writing requires more significant amendment to improve the clarity and I have left it up to the authors to determine the most appropriate changes.

Thank you very much indeed! We have considered all your suggestions and done our best to improve the clarity throughout.

Line 452: R2 - As worded this is confusing.  Please reword so that the significant change and its direct is clear- e.g., there was a clear decrease in ...  Ideally, there was a statistically significant decrease in....

The sentence “There was a clear change (decrease from the playback on) in the number of stereotypic behaviors observed according to periods (Friedman chi-squared = 6.9, df = 2, p-value = 0.03) with a clear decrease between the PRE and POST periods …” was changed for: “There was a statistically significant reduction in the number of stereotypic behaviours (Friedman's chi-square = 6.9, df = 2, p-value = 0.03), particularly between the PRE and POST periods …”.(now lines 478-486)

Ligne 457: As written this is difficult to follow- please describe the results- e.g., Horses exhibited fewer agitation behaviours during the POST period than the during the PRE-period, (results).  This makes it much easier for the reader to understand what the statistics are saying.  Also, direct the reader to the graphs so they can compare the data with the statistics.

The sentence : “Horses exhibited fewer agitation behaviors (Friedman chi-squared = 7.1, df = 2, p= 0.03 ; PRE / PL : Wilcoxon V=54, BH correction : p=0.02 ; PRE / POST : Wilcoxon V=42, BH correction : p=0.04) and vacuum chewing during the POST period than during the PRE period (Friedman chi-squared = 10.2, df = 2, p= 0.006; PRE / POST : Wilcoxon V=70, BH correction : p=0.04)” was changed for: “Horses showed fewer agitation behaviors and vacuum chewing during the POST period than during the PRE period (agitation: Friedman chi-square = 7.1, df = 2, p= 0.03; PRE / POST: Wilcoxon V=42, BH correction: p=0.04; vacuum chewing: Friedman chi-square = 10.2, df = 2, p= 0.006; PRE / POST: Wilcoxon V=70, BH correction: p=0.04) (fig 3b,c).” (now lines 482-485)

The reference list is extensive and relevant. Some additional references are required as per the annotated copy of the paper

The references added in connection with the changes made have all been added at the end of the list of references.

Round 2

Reviewer 1 Report

I thank the authors for the thoroughness of their response to issues raised and for the changes made. These improve the objectivity and rigour of the paper and reassure readers of the efforts made to perform a study with defensible results supported by appropriate methodology. I remain concerned that the manner of presentation of these interesting and valuable findings not unnecessarily raise concerns in the reader's mind as to the validity of the findings and their interpretation.

There are one or two items to which the authors may wish to apply a little more time. I offer these with the intention that they may increase the impact of the study and not as points of disagreement. I will leave it to the authors and the editor to determine whether any attention is required to the following - I do not need to see the paper again.

The revisions have increased the length of the paper, an unfortunate outcome. One element of the introduction might be modified in the interests of both length and clarity, the frequent reference to A4 440 vs 432. This tuning debate will likely never be resolved until some very extensive research is performed. In the context of what the paper intends to do, it would become relevant if the subject of discussion was the actual playing of music. In the present study, reference is made to pure frequencies used in other experiments, while much of the literature cited refers to responses to specific frequencies - which will be the same whether the musical standard is 440 or 432. It would be interesting to see which tuning standard horses prefer, but that's not the question here. Reference to the standard is confusing when other reference is to specific sound frequencies. 

It might be sufficient simply to observe that the tuning debate exists with regard to the tuning used in studies testing the response of animals to music, and that this issue needs to be standardised if results are to be interpreted. The debate is not relevant to this study because, apparently, the horses were exposed to pure sound and not music. The level of detail covered would be better placed in a review paper. Indeed, the issues raised would need to be considered by anybody inspired by this paper to conduct their own research. Such a review in the animal literature, therefore, would be extremely valuable and potentially a much-cited reference source. Conversely, a potentially interested party might be put off by the apparent complexity of the issues. These issues are compounded by the contents of the paragraph starting at line 183, when you refer frequently to the use of specific frequencies and pure tones.

The title is still problematic. Scientific objectivity requires we just study impacts using the null hypothesis and describe the findings - your title implies that positive impacts are a given as you have previously studied them and found them to be positive. A more accurate title would be "A preliminary study of the impacts of selected acoustic frequencies on behavioural and physiological welfare indicators in Thoroughbred racehorses". If you are determined to indicate your positive findings in the title, then the title becomes something like "Selected acoustic frequencies have a positive impact on behavioural and physiological welfare indicators in Thoroughbred racehorses", although this will of course sharpen the reader's focus on the fact your study was limited by confidentiality issues.

The authors have not responded to my questions concerning laboratory standards and use of normal values from a different laboratory.

Line 25. "parameters"

Line 26, "and potentially experienced a better physical recovery"

Line 89, "Seasons"

Line 167, please be consistent in citation method. Place only the citation number in square brackets, not text. Use of the abbreviation "e.g." isn't really necessary - the citation number essentially says the same thing. 

Line 320, "stimulus of human caretakers…"

Line 331, do you mean "Heart rate measures do not really help with durable states as their correlation with stereotypic behaviours…"?

Figure 4, figure appears to be corrupted in the copy I was sent and partially overlies section 3.4. The unit on the Y axis is not indicated - I assume this is minutes?

Line 548, "large" or "big" enough?

Line 583, "non-audible", not fully dealt with in your response, but it might be helpful if you made it clear that you are referring (if you are) to the fact that the sounds were not audible to the humans in the environment.

Line 613, "… which is a multivariable stimulation…"

Line 708,, "… beginnings of our understanding…", "… frontiers of our understanding…", "… outset of our understanding…"?

Line 749. I wonder if your use of the expression "carrier frequency" is not in error. I defined this term in my review, but if you are exclusively referring to the frequency standard for tuning musical instruments, then that is the term you should be using. See my comments above.

Line 769, "… but also the way the whole body…"

Limitations. These should be stated without citations, but simply listed directly and clearly. Otherwise, the limitation section starts to become a discussion. Points that have, or could have, been raised in the discussion should not be repeated in Limitations.

Conclusions. The following observations indicate how you might state your conclusions to avoid overstatement and increase objectivity - they are not intended as a rewrite!

Line 812. It is not necessary to include a description of the study, which should be clear by this point.

Line 814. Your study was not designed to answer this question. You had no controls. The most you can say is that the treatment reduced the frequency of behaviours associated with stress and increased behaviours associated with relaxation.

Line 819. "These findings open new lines for future research involving designed, controlled studies with larger population sizes and examination of the effects of individual frequencies."

Line 821. "This would promote improved welfare, which could in turn lead to improved performance. It would also help us to better understand how non--audible stimuli might have the observed effects on  behaviours." Reference to "major effects" is overstatement, especially from a non-controlled study with limitations.

Appendix A. Please define "Nb" in the legend

Author Response

I thank the authors for the thoroughness of their response to issues raised and for the changes made. These improve the objectivity and rigour of the paper and reassure readers of the efforts made to perform a study with defensible results supported by appropriate methodology. I remain concerned that the manner of presentation of these interesting and valuable findings not unnecessarily raise concerns in the reader's mind as to the validity of the findings and their interpretation.

Thank you for this positive statement.

With regard to the presentation of the results, we had already lowered their interpretation at the time of the first revision, and we had also increased the limitation part as requested. This seemed to satisfy reviewer 2, who made no comment on it.

Nevertheless, in order to go further, we have, in this second revision, again carefully studied all the comments and suggestions you have made below.

There are one or two items to which the authors may wish to apply a little more time. I offer these with the intention that they may increase the impact of the study and not as points of disagreement. I will leave it to the authors and the editor to determine whether any attention is required to the following - I do not need to see the paper again.

Thanks for these propositions.

The revisions have increased the length of the paper, an unfortunate outcome. One element of the introduction might be modified in the interests of both length and clarity, the frequent reference to A4 440 vs 432. This tuning debate will likely never be resolved until some very extensive research is performed. In the context of what the paper intends to do, it would become relevant if the subject of discussion was the actual playing of music. In the present study, reference is made to pure frequencies used in other experiments, while much of the literature cited refers to responses to specific frequencies - which will be the same whether the musical standard is 440 or 432. It would be interesting to see which tuning standard horses prefer, but that's not the question here. Reference to the standard is confusing when other reference is to specific sound frequencies. 

In the introduction, we wanted to be as exhaustive as possible in order to take stock of the existing literature, which focuses mainly on the effects of music. Even if this is not the primary objective of this paper (study of pure frequencies), it shows the reader the innovative aspect of this manuscript and helps to explain the lack of information in order to best interpret our results.

It might be sufficient simply to observe that the tuning debate exists with regard to the tuning used in studies testing the response of animals to music, and that this issue needs to be standardised if results are to be interpreted. The debate is not relevant to this study because, apparently, the horses were exposed to pure sound and not music. The level of detail covered would be better placed in a review paper. Indeed, the issues raised would need to be considered by anybody inspired by this paper to conduct their own research. Such a review in the animal literature, therefore, would be extremely valuable and potentially a much-cited reference source. Conversely, a potentially interested party might be put off by the apparent complexity of the issues. These issues are compounded by the contents of the paragraph starting at line 183, when you refer frequently to the use of specific frequencies and pure tones.

As previously stated, we wanted to be as exhaustive as possible in the introduction in order to take stock of the existing literature, which focuses mainly on the effects of music, even if this is not the primary objective of this document. In addition, at the time of the last revision

In addition, doing a separate revision was a good idea, and we considered it.  However, as we responded in the last review, after discussion with the Animals board, it became apparent that it would be costly and difficult to include both articles in this issue.

The title is still problematic. Scientific objectivity requires we just study impacts using the null hypothesis and describe the findings - your title implies that positive impacts are a given as you have previously studied them and found them to be positive. A more accurate title would be "A preliminary study of the impacts of selected acoustic frequencies on behavioural and physiological welfare indicators in Thoroughbred racehorses". If you are determined to indicate your positive findings in the title, then the title becomes something like "Selected acoustic frequencies have a positive impact on behavioural and physiological welfare indicators in Thoroughbred racehorses", although this will of course sharpen the reader's focus on the fact your study was limited by confidentiality issues.

We have replaced the title “A first study on the positive impact of selected acoustic frequencies on behavioural and physiological welfare indicators in thoroughbred racehorses” by “Selected acoustic frequencies have a positive impact on behavioural and physiological welfare indicators in Thoroughbred racehorses”

The authors have not responded to my questions concerning laboratory standards and use of normal values from a different laboratory.

The standards of the laboratory that carried out the analyses are not racehorse standards, which is why we have based ourselves on the standards of another laboratory specialising in thoroughbreds.

We have added this to the manuscript (line 422-424).

Line 25. "parameters"

We have added a “s” to parameter (line 23)

Line 26, "and potentially experienced a better physical recovery"

We have replaced the sentence “Overall thus, the animals appeared much quieter and had probably a better physical recovery.” by “Overall thus, the animals appeared much quieter and potentially experienced a better physical recovery.” (line 24-25)

Line 89, "Seasons"

We put a capital “S” to Seasons (line 88)

Line 167, please be consistent in citation method. Place only the citation number in square brackets, not text. Use of the abbreviation "e.g." isn't really necessary - the citation number essentially says the same thing. 

We have only placed references in square brackets. So we've changed that “Moreover, there is evidence of the influence of sound vibrations in vertebrate and non-vertebrate species that lack an auditory system [e.g gilthead sea bream: 19; snails: 61]” by “Moreover, there is evidence of the influence of sound vibrations in vertebrate and non-vertebrate species that lack an auditory system (gilthead sea bream: [19] ; snails: [61])”. (line 166)

This has been changed throughout the manuscript.

Line 320, "stimulus of human caretakers…"

We've not taken this into account and kept our original sentence "with possible effects of the stimulus on human caretakers/riders who arrived at 6:00 am [91]." and not " with possible effects of the stimulus of human caretakers/riders who arrived at 6:00 am [91]." because we want to show that the stimulus can also have an influence on human behavior (as shown in other papers).

Line 331, do you mean "Heart rate measures do not really help with durable states as their correlation with stereotypic behaviours…"?

Yes, we mean that.

We have replaced the sentence “Heart rate measures do not really help with durable states as it correlates with stereotypic behaviours or pain is controversial [e.g. 94, 95, 96] and changes in welfare states are not necessarily associated with heart rate changes [e.g. 97].” by “Heart rate measures do not really help with durable states as their correlation with stereotypic behaviours or pain is controversial [e.g. 94, 95, 96] and changes in welfare states are not necessarily associated with heart rate changes [e.g. 97]” (line 329-330)

Figure 4, figure appears to be corrupted in the copy I was sent and partially overlies section 3.4. The unit on the Y axis is not indicated - I assume this is minutes?

However, we have no layout problems with our version of the manuscript.

Yes, this is minutes, it has been added to the figure 4.

Line 548, "large" or "big" enough?

This is an omission from the last revision. We are keeping "large" and removing "big".

So we have replaced the sentence : “These decreases were not large big enough to have levels…” by “These decreases were not large enough to have levels…”.  (line 532)

Line 583, "non-audible", not fully dealt with in your response, but it might be helpful if you made it clear that you are referring (if you are) to the fact that the sounds were not audible to the humans in the environment.

We have added a clarification here.

We have replaced the sentence “This study, where 12 racehorses were submitted twice daily to the playback of an array of non-audible pure tone frequencies during three weeks (i.e. a total of 21 hours of observation) in their home stalls, is to our knowledge unique both in the system used and in the extent of the observed major effects on both behavior and physiology.” by “This study, where 12 racehorses were submitted twice daily to the playback of an array of pure tone frequencies non-audible  to the humans in the environment) during three weeks (i.e. a total of 21 hours of observation) in their home stalls, is to our knowledge unique both in the system used and in the extent of the observed major effects on both behavior and physiology.” (line 556).

Line 613, "… which is a multivariable stimulation…"

We have changed “… which is a that is a multivariable stimulation…” by “…which is a multivariable stimulation…”. (line 585)

Line 708,, "… beginnings of our understanding…", "… frontiers of our understanding…", "… outset of our understanding…"?

We have replaced the sentence “We are clearly at the premises of understanding what exactly explains the observed positive or negative effects of sound stimulation” by “We are clearly at the beginnings of our understanding of what exactly explains the observed positive or negative effects of sound stimulation” (line 709)

Line 749. I wonder if your use of the expression "carrier frequency" is not in error. I defined this term in my review, but if you are exclusively referring to the frequency standard for tuning musical instruments, then that is the term you should be using. See my comments above.

We've taken your previous comments on board and responded point by point. Thank you.

Line 769, "… but also the way the whole body…"

We replaced the sentence “A greater understanding of not only the auditory system but also the whole body processes the information is needed” by “A greater understanding of not only the auditory system but also the way the whole body processes the information is needed”. (line 736)

Limitations. These should be stated without citations, but simply listed directly and clearly. Otherwise, the limitation section starts to become a discussion. Points that have, or could have, been raised in the discussion should not be repeated in Limitations.

We have taken this into consideration

Conclusions. The following observations indicate how you might state your conclusions to avoid overstatement and increase objectivity - they are not intended as a rewrite!

Thank you for your suggestions. We'll respond below.

Line 812. It is not necessary to include a description of the study, which should be clear by this point.

We have therefore reworded this sentence by removing : “where the behavior, physiology and welfare of racehorses were studied before, during and after a 3-week period of playback of a pure-frequencies commercial non-audible acoustic stimulus”.

So, we replaced the sentence “The results of this study where the behavior, physiology and welfare of racehorses were studied before, during and after a 3-week period of playback of a pure-frequencies commercial non-audible acoustic stimulus…” by “The results of this study…”. (line 781)

Line 814. Your study was not designed to answer this question. You had no controls. The most you can say is that the treatment reduced the frequency of behaviours associated with stress and increased behaviours associated with relaxation.

The aim of this first concluding point was to show in a general way the state the horses were in before the sound was broadcast, i.e. without the need for a control group.

However, as this seemed confusing, we have revised the sentence to avoid generalising it to all racehorses. So, we replaced the sentence “1) confirm that racehorses at training do exhibit a compromised welfare as pre-playback, with most horses exhibiting stereotypic behaviors, agitation and high frequencies of yawning and vacuum chewing, as well as abnormally high levels of lymphocytes and MCV (considered as an indicator of training stress)” by “1) show that, before the sounds were broadcasted, the observed racehorses showed compromised welfare, with most of them exhibiting stereotypic behaviors, agitation and/or high frequencies of yawning and vacuum chewing, as well as abnormally high levels of lymphocytes and MCV (considered as an indicator of training stress)”. Line (781-785)

Line 819. "These findings open new lines for future research involving designed, controlled studies with larger population sizes and examination of the effects of individual frequencies."

We replaced the sentence “These findings open new lines for future research, ideally on larger samples of animals and individualized frequencies.” by “These findings open new lines for future research involving studies with larger population sizes and control horses as well as examination of the effects of individual frequencies”. (line 787-789)

Line 821. "This would promote improved welfare, which could in turn lead to improved performance. It would also help us to better understand how non--audible stimuli might have the observed effects on behaviours." Reference to "major effects" is overstatement, especially from a non-controlled study with limitations.

We replaced the sentence “This would allow both to improve the welfare conditions and possibly performances in racehorses but also to better understand how non-audible stimuli can have such major effects on their mental and physical health.” by “This would promote improved welfare, which could in turn lead to improved performance. It would also help us to better understand how non-audible stimuli might have the observed effects on   their mental and physical health.” (line 789-791)

Appendix A. Please define "Nb" in the legend

We have added "Nb= Number" to the legend of Appendix A

Reviewer 2 Report

The authors are to be commended for taking such a conscientious and detailed approach to addressing the reviewer's concerns.  These concerns have been addressed in full and the paper is now ready for publication subject to final edit/spell check.  This is an innovative and interesting paper and it makes an important contribution to the literature.

Author Response

The authors are to be commended for taking such a conscientious and detailed approach to addressing the reviewer's concerns.  These concerns have been addressed in full and the paper is now ready for publication subject to final edit/spell check.  This is an innovative and interesting paper and it makes an important contribution to the literature.

Thank you very much for this positive statement.